# Surface circulation in the Gulf of Thailand from remotely sensed observations: seasonal and interannual timescales

Arachaporn Anutaliya[1]

[1]Institute of Marine Science, Burapha University, Chonburi, Thailand

**Correspondence:** Arachaporn Anutaliya (arachaporn.an@go.buu.ac.th)

**Abstract.** The Gulf of Thailand (GoT), a shallow semi-enclosed basin located in the western equatorial Pacific, undergoes much wind variabilities on both seasonal and interannual timescales that produce complex surface circulation. The local Ekman pumping modifies sea level in the northern GoT, while remote wind forcing influences sea level variability at the GoT western boundary, potentially through the coastal trapped Kelvin waves. The importance of the Ekman current on ageostrophic current is also important; the stronger influence of the Ekman current is found toward the southern part of the GoT. The GoT circulation reverses its direction seasonally following the monsoon wind reversal which is well-captured by the most dominant complex empirical orthogonal function explaining 28% of the total circulation variance. During the monsoon transition, a strong meridional current along the western boundary that connects to the flow at the GoT southeastern entrance is observed. This implies high exchange between the GoT and the South China Sea, and thus modification of the GoT water. On the interannual timescale, the GoT circulation is directly impacted by both El Niño Southern Oscillation (ENSO) and the Indian Ocean Dipole (IOD). Interestingly, the two climate modes have different spatial influences on the GoT circulation. The IOD dominates the interannual current along the GoT western boundary and the southern boundary of the observing domain (8° N), while the ENSO correlates with that in the interior. The results highlight the complex circulation pattern as being contributed by different dynamics over each region of the GoT.

## 1   Introduction

The Gulf of Thailand (GoT), located at 8°-14° N, 99° - 105° E (Figure 1), is a shallow semi-enclosed basin with an average depth of 40 m that is largely influenced by winds on both seasonal and interannual timescales. On seasonal timescale, the Asian monsoon winds prevail producing the wet season over southeast Asia from approximately May to August (southwest monsoon) and the dry season from November to February (northeast monsoon). Although the GoT circulation is also heavily dependent on inflows from the SCS, e.g., along the eastern coast of Malaysia located to the south of ∼6.7° N and around the southern coast of Vietnam, these currents are mainly driven by the monsoon winds (e.g. Wyrtki, 1961; Akhir, 2012). The extreme seasonal wind and precipitation conditions influence the circulation pattern, physical seawater properties (e.g. salinity, density, hence the thermohaline circulation; Yanagi and Takao, 1998; Yanagi et al., 2001; Buranapratheprat et al., 2002, 2008), and nutrients loadings from rivers (e.g. nitrate, phosphate, and ammonia; Suvapepun, 1991; Sriwoon et al., 2008). Also, the location of the GoT which is to the west of the South China Sea (SCS) in the equatorial western Pacific Ocean provides a unique opportunity

to observe the influence of both large-scale climate modes in the Pacific Ocean (El Niño Southern Oscillation: ENSO) and the Indian Ocean (Indian Ocean Dipole: IOD; Saji et al., 1999) on the interannual circulation.

Previous observational and numerical studies show that circulation in the Gulf of Thailand varies seasonally (e.g., Yanagi et al., 2001; Buranapratheprat et al., 2008; Saramul, 2017; Buranapratheprat et al., 2002; Aschariyaphotha et al., 2008; Saramul and Ezer, 2014). A series of 6 hydrographic cruises over October 2003 – July 2005 in the upper GoT (uGoT; north of 12.5° N) suggests an overall cyclonic circulation during the northeast monsoon (Buranapratheprat et al., 2008) in agreement with numerical studies that account for tidal forcing, bottom friction, and river runoffs (Buranapratheprat et al., 2002; Saramul and Ezer, 2014). The numerical studies also suggest the dominance of contrasting anticyclonic circulation in the uGoT during the southwest monsoon. Still, both numerical simulations are forced by spatially uniform reanalysis wind products which likely do not represent the actual wind field over the region (Yanagi and Takao, 1998). A study based on numerical simulation forced by spatially varying wind (Buranapratheprat et al., 2006) emphasizes the importance of both zonal and meridional wind gradients on the circulation over the uGoT. For example, the development of an anticyclonic circulation to the north of 13° N during the southwest monsoon, observed by Buranapratheprat et al. (2002) and Saramul and Ezer (2014), is highly dependent on the intensity of wind at the south or east of the uGoT. Results from hydrographic surveys in May 2004 and July 2005 do not show a clear dominant circulation pattern during this period (Buranapratheprat et al., 2008). Fine-spatial-resolution coastal radar of the monthly-mean surface current during both the southwest (June 2015) and northeast monsoon (February 2015) reveals a complex circulation pattern in the uGoT, although the circulations during the two seasons are not distinctly different (Saramul, 2017). The circulation based on the coastal radar suggests an overall cyclonic circulation in the northern part of the uGoT (north of 12.8°-12.9° N) and an anticyclonic circulation in the southern part (south of 12.8°-12.9° N) during both monsoon seasons.

South of 12.5° N, various observational and numerical studies were conducted (Wyrtki, 1961; Yanagi and Takao, 1998; Aschariyaphotha et al., 2008; Sojisuporn et al., 2010); however, the findings are not quite consistent due to the different studied periods and spatial resolution being considered. During the southwest monsoon, altimetry-based observation over the 1995 - 2001 period (Sojisuporn et al., 2010) shows that the geostrophic circulation intensifies at the rim of the GoT with a strong southward current within 1° of the GoT western boundary and westward current to the south of the uGoT yielding a cyclonic circulation. At the southeastern entrance, satellite altimetry indicates an outflow into the SCS (Sojisuporn et al., 2010). Numerical simulation assimilating measurements from the NAGA expedition in 1959-1960 (Yanagi and Takao, 1998) shows similar results, except the presence of strong northwestward flow in the mid-basin which yields a cyclonic circulation to its west and an anticyclonic circulation to its east. In contrast, findings from Princeton Ocean Model (Aschariyaphotha et al., 2008) indicate a strong southeastward flow in the mid-basin and the dominance of an anticyclonic circulation over the GoT with an outflow at the southeastern entrance during the southwest monsoon. Note that the study by Aschariyaphotha et al. (2008) allows inflow and outflow from the lateral boundaries while that by Yanagi and Takao (1998) does not which could contribute to the discrepancy. Still, the circulation pattern found by Aschariyaphotha et al. (2008) resembles the GoT surface velocity surveyed during

the NAGA expedition (Wyrtki, 1961).

During the northeast monsoon, the altimetric observations and numerical simulations suggest the dominance of an anticyclonic circulation at the rim of the GoT and an inflow at the southeastern entrance (Yanagi and Takao, 1998; Aschariyaphotha et al., 2008; Sojisuporn et al., 2010). Circulation in the GoT interior is quite complex and findings from the studies do not necessarily agree. The NAGA expedition, however, shows that cyclonic circulation prevails during the northeast monsoon (Wyrtki, 1961). Still, an inflow is present at the southeastern entrance in agreement with the observational and numerical studies. The inflow is found to reach the bottom at the GoT western boundary during the spring monsoon transition as a hydrographic survey shows the presence of cold and saline water originating in the SCS at the region (Yanagi et al., 2001).

Although previous studies have recognized the role of monsoon winds on seasonal variability of the GoT circulation (e.g., Yanagi and Takao, 1998; Aschariyaphotha et al., 2008), the associated dynamics are not well understood. Therefore, this study aims to examine the seasonal variability of surface circulation in the GoT and the associated dynamics by investigating the influence of geostrophic current and wind-driven Ekman current using remotely-sensed observations. The mechanisms that set up the geostrophic flow will also be discussed. In addition, interannual variability of the GoT circulation will be examined to understand the effect of ENSO and the IOD on the circulation pattern.

## 2   Datasets

To examine circulation pattern in the GoT, the gridded Ocean Surface Currents Analyses Real-time product (OSCAR; Bonjean and Lagerloef, 2002) between 8° and 14° N, 99° and 105° E is considered (Figure 1). The product is calculated based on satellite sea surface height, wind, and water temperature from both remotely-sensed and in-situ measurements, e.g. drifters, moored, and shipboard measurements, etc. The resulting current is an average in the upper 30 m of the water column. Therefore, OSCAR current represents the total current (sum of the geostrophic and ageostrophic currents) over the GoT. The gridded OSCAR product has a resolution of 1/3° (36-37 km in the GoT) with a temporal resolution of 5 days available from 1992 to 2020. To validate the OSCAR velocity over the GoT, the monthly average velocity maps in February 2015 and June 2015 are compared to tide-removed surface currents from high-frequency radar system shown in Saramul (2017). Generally, OSCAR velocity exhibits a similar circulation pattern to the coastal-radar velocity, particularly in February 2015. Still, a much more complex circulation is observed in the coastal-radar velocity due to its much finer spatial resolution. The difference between OSCAR velocity and high-frequency coastal-radar velocity is the largest in the uGoT; as the region is quite small and shallow (Figure 1), the spatial resolution provided by the OSCAR products might not be sufficient to resolve the circulation there.

The gridded all-satellite merged absolute dynamic topography (ADT; $\eta$) product is used to calculate geostrophic current ($u_g$ denotes zonal velocity and $v_g$ denotes meridional velocity) in the GoT and the associated mechanisms:

$$u_g = -\frac{g}{f}\frac{\partial \eta}{\partial y}, \text{ and} \tag{1}$$

95

$$v_g = \frac{g}{f}\frac{\partial \eta}{\partial x}, \tag{2}$$

where $g$ represents the gravitational acceleration, $f$ is Coriolis parameter, $y$ is distance in the meridional direction, and $x$ is distance in the zonal direction. The product interpolated daily on a 1/4° grid (27-28 km in the GoT) available over the 1993 − 2020 period (Ducet et al., 2000). As the satellite altimetry used here is in the coastal region (Figure 1), sea surface level data from 7 tide gauge stations in the GoT: Fort Phrachula Chomklao (FP), Ko Lak (KL), Ko Mattaphon (KM), Huahin (HH), Ko Prap (KP), Laem Sing (LS), and Ko Sichang (KS) (Holgate et al., 2013; PSMSL, 2019), are used to validate the satellite-derived ADT. The comparisons show strong correlation between the fluctuation of satellite ADT and the tide gauge sea level over the 2014-2019 period with correlation coefficients ranging from 0.69 to 0.94 (Figure 1). In addition, the ADT shows distinct seasonal cycle consistent with water level measurements from tide gauges (Trisirisatayawong et al., 2011). The weaker correlation found at FP is largely due to land subsidence caused by the high volume of groundwater extraction; the land subsidence introduces a rise in the tide gauge sea level (Adebisi et al., 2021; Jaroenongard et al., 2021). Still, the lower correlation could also be contributed by the tide gauge's location which is ~20 km inland from the available gridded satellite ADT, or the error that the ADT might have at that location.

To examine the effect of wind-driven Ekman current on the GoT circulation, the gridded surface vector winds Version 2 Cross-Calibrated Multi-Platform (CCMPv2) obtained from Remote Sensing Systems are used. The CCMPv2 wind product, available from July 1987 to December 2019, has a resolution of 1/4° with a temporal resolution of 6 hours (Wentz et al., 2015). The Ekman current ($u_e$ and $v_e$) at each depth ($z$) is calculated following Alberty et al. (2019):

$$u_e(z) = \frac{\sqrt{2}}{fd}e^{z/d}\left[\tau^x\cos\left(\frac{z}{d} - \frac{\pi}{4}\right) - \tau^y\sin\left(\frac{z}{d} - \frac{\pi}{4}\right)\right], \tag{3}$$

$$v_e(z) = \frac{\sqrt{2}}{fd}e^{z/d}\left[\tau^x\sin\left(\frac{z}{d} - \frac{\pi}{4}\right) + \tau^y\cos\left(\frac{z}{d} - \frac{\pi}{4}\right)\right], \tag{4}$$

where $\tau$ is wind stress, $d$ is thickness of the surface Ekman layer defined as $\sqrt{\frac{2A}{|f|}}$ with $A$ being a function of wind speed ($|\mathbf{U}|$; $A = 8 \times 10^{-5}|\mathbf{U}|^{2.2}$). In addition, wind stress curl ($\nabla \times \tau = \frac{\partial \tau^y}{\partial x} - \frac{\partial \tau^x}{\partial y}$) is also calculated.

The weekly sea surface temperature averaged over the Niño 3.4 box (hereafter referred to as Niño3.4) provided by the National Oceanic and Atmospheric Administration (NOAA) is used to indicate ENSO conditions (Trenberth, 1997). To assess

the IOD conditions, the Dipole Mode Index (DMI) is used. The weekly DMI calculated from sea surface temperature in the tropical Indian Ocean is calculated and provided by the NOAA/ Earth System Research Laboratory (Saji et al., 1999; Black et al., 2003).

## 3 Methodology

### 3.1 Complex empirical orthogonal function

To determine the dominant pattern and the associated temporal variation of the surface current in the GoT, the complex empirical orthogonal function (CEOF) is utilized. The CEOF is similar to the empirical orthogonal function (EOF) which is suitable for analysis of a dataset with both spatial and temporal variation (e.g., Weare et al., 1976; North et al., 1982). The EOF technique decomposes the data matrix that has its mean removed ($\mathbf{X}$) into orthogonal EOF modes ($U$) that display spatial patterns. Each mode corresponds to a time series known as the principal component (PC) that demonstrates the temporal variation of that EOF mode; the PC identifies when and how intense each EOF pattern occurs. The collection of the PCs forms an orthogonal matrix ($V$). The matrix decomposition is done as follows:

$$\mathbf{X} = USV^T, \tag{5}$$

where the superscript $T$ denotes the transposition of a matrix. $S$ contains the magnitude of a linear transformation; it designates the intensity of each EOF mode. The fractions of variance explained by EOF modes are different and variance of the $i$-th mode is calculated as $\frac{S_{i,i}^2}{\sum_j S_{j,j}^2}$; the first EOF mode shows the most dominant pattern and the subsequent modes account for a smaller fraction of the variance by the mathematical construction. When the technique is applied to vector quantities, e.g., velocity, the CEOF is often adopted, where each vector is transformed into a complex number (e.g., Kundu and Allen, 1976; Klinck, 1985). In this study, the velocity vector with the time-mean removed ($\mathbf{u}$) is decomposed as

$$\mathbf{u} = u + iv, \tag{6}$$

where $u$ is the zonal velocity, $v$ is the meridional velocity, and $i$ is $\sqrt{-1}$. Applying the same EOF technique (Eq. 5) to the complex number, the resultant PC is complex where its magnitude represents the temporal fluctuation of the corresponding CEOF. The phase, calculated as the arctan of the imaginary part divided by the real part, represents the direction that the CEOF mode has to rotate (positive clockwise).

### 3.2 Complex correlation

To understand the relationship between two vector time series, e.g., the surface velocity at two different locations, a complex correlation analysis (Kundu, 1976) can be applied to time series of complex numbers ($\mathbf{u}(t)$) constructed as shown in Eq. 6. The complex correlation coefficient ($R$) is computed as

$$R = \frac{\mathbf{u}(t)_1 \mathbf{u}(t)_2^*}{\sqrt{(\mathbf{u}(t)_1 \mathbf{u}(t)_1^*)(\mathbf{u}(t)_2 \mathbf{u}(t)_2^*)}}, \tag{7}$$

where * denotes complex conjugate. The resultant $R$ is a complex number where its magnitude describes how the magnitude of the two time series covary. The phase of $R$, computed as the arctan of the imaginary component divided by the real component, describes the angle between the two vector time series in order to achieve the highest correlation.

## 4 Circulation in the Gulf of Thailand

The mean and variance of OSCAR surface velocity are calculated over the 2014 - 2019 period (Figure 2). Strong mean flow is observed near the northern and western boundaries of the GoT and at the southeastern entrance. The mean current is northward along the western boundary to the south of 12° N and southwestward at the southeastern entrance. The mean circulation in the GoT interior consists of a few weak eddies. The mean circulation pattern from OSCAR products generally agrees with the satellite-derived geostrophic current (color contour in Figure 2a), except in the uGoT where OSCAR products are present

at only six locations. Also, as the uGoT is shallow and enclosed by land on the western, northern, and eastern sides (Figure 1), OSCAR products over the region could contain a substantial error. Thus, discussion regarding OSCAR velocity over the uGoT will be omitted. Large variance of the surface circulation is found along the western boundary of the GoT, approximately between 9.5° and 11.5° N, with most of the variance associated with meridional velocity (Figure 2b). Variance of the ADT is also high along the western boundary indicating the influence of geostrophic flow (Eq. 1 and 2), particularly between 9° and

10.5° N. At the southeastern entrance, high variance is observed in both OSCAR velocity and ADT with the highest velocity variance observed at the southeastern part of the observing domain and highest ADT variance observe a bit farther north, at ~9° N.

To understand the variability of the GoT circulation, the mean surface velocity (quivers in Figure 2a) is removed from the

OSCAR current to calculate Complex Empirical Orthogonal Function (CEOF; see Methodology section). The first few CEOF modes explain 28%, 14%, 10%, and 7.4% of the surface current variance, respectively. This study only selects the first two modes to represent the dominating GoT circulation patterns (Figure 3). The first CEOF mode describes a strong southward flow along the western boundary and a strong southeastward flow at the southeastern entrance during the southwest monsoon and fall monsoon transition (Figure 3a-c). In the GoT interior, an anticyclonic circulation centered at ~10.5° N, 101.5° E

is present during these seasons. During the northeast monsoon and spring monsoon transition, the circulation reverses its direction. The circulation pattern is weaker during the monsoon transition compared to that during the monsoon seasons and it is often weak or absent at the end of the spring monsoon transition/ beginning of the southwest monsoon and at the end of the fall monsoon transition/ beginning of the northeast monsoon (Figure 3b). The second CEOF mode highlights strong flow along the western boundary of the GoT with the strongest flow in the northern part and weaker flow toward the central and

lower GoT resembling the pattern of the circulation variance (Figure 2b, 3d). Circulation in the GoT interior and along the eastern boundary is generally weak, except at the southeastern entrance. Although the phase does not exhibit a distinct pattern (Figure 3f), the negative phase indicating southward flow is found during the southwest monsoon and into the fall monsoon transition (July – October) of 2016 – 2019 which could be associated with river runoffs (Aschariyaphotha and Wongwises,

2012). Southward flow is observed along the western boundary with the strongest current in the northern part of the GoT and a weak southwestward flow is present at the southeastern entrance. In addition, a positive phase is found during the spring monsoon transition over the observing period except in 2016; this indicates northward flow along the western boundary and northeastward flow at the southeastern entrance which could be associated with the SCS inflow (Yanagi et al., 2001; Aschariyaphotha et al., 2008).

## 4.1 Seasonal circulation in the Gulf of Thailand

### 4.1.1 Overall description

As suggested by the dominating circulation pattern calculated based on the CEOF, more than one-quarter of the variance in the GoT current can be simply explained by an annually reversing circulation that follows the monsoon seasons (Figure 3a-c). Thus, the monthly mean current over the 2014 – 2019 period is calculated; the monthly current reveals a circulation pattern generally consistent with that shown by CEOF1 (Figure 4a, c, e, g), particularly in June (representing the southwest monsoon) and December (representing the northeast monsoon). The pattern describes the circulation with strong current at three main regions which are the western boundary of the GoT, the interior of the GoT, and the southeastern entrance of the GoT. In the interior of the GoT, an anticyclonic circulation is present (centered at ∼10.5° N, 101.5° E) during the southwest monsoon (Figure 4c, d). A strong southward current is observed along the western boundary and a strong southeastward current is observed at the southeastern entrance suggesting an outflow into the SCS at the surface. The circulation pattern generally reverses its direction during the northeast monsoon, consistent with a previous surface current observation (Saramul, 2017). The monthly mean surface current hints at the connection between the currents along the western boundary and that at the southeastern entrance (Figure 4a, c, e, g). Thus, a complex correlation (Eq. 7) between the current along the western boundary and that over the GoT is performed to understand the dynamics associated with the strong western boundary current particularly if it is related to the current at the southeastern entrance. On timescales longer than 30 days, currents along the southern boundary of the domain (south of 8.5° N) give a higher correlation to the western boundary current (9.0° - 11.5° N, 99.5° - 100.2° E) compared to the rest of the GoT (Figure 5); the correlation is higher than 0.25 with the highest value of 0.57 at the entrance. The correlation between the western boundary current and that along the southern boundary of the domain is significant with 95% confidence as determined by a non-parametric method (Sprent and Smeeton, 2007) where correlation coefficients are computed repeatedly (5000 times) using both of the time series that are randomly rearranged. The significant correlation strongly suggests a connection between the GoT western boundary current and the GoT inflow/outflow at the southeastern entrance through a passage to the south of ∼8.5° N.

During the spring monsoon transition (represented by March), the current resembles that during the northeast monsoon; a westward flow at the southeastern entrance is observed. The cyclonic circulation in the GoT interior is still present, but weak. However, the northward flow along the western boundary is stronger and wider compared to that during the northeast monsoon; the northward current extends more than 80 km offshore to ∼100° E (Figure 4a, d). Similarly, the GoT circulation in September,

representing the fall monsoon transition, resembles that during the southwest monsoon despite the weak anticyclonic circulation in the interior. The circulation pattern during the monsoon transitions shows the dominant circulation pattern captured by both CEOF1 and CEOF2 (Figure 3) reflecting the influence of monsoon winds and of the current connecting to the SCS (Figure 5).

### 4.1.2 Geostrophic and ageostrophic component

Satellite altimetry is used to estimate geostrophic components of the surface circulation over the GoT. Although the altimetry may include short-period contribution during the satellite over-pass, geostrophic velocity calculated from the altimetry is found to be reasonably close to the observation (Yu et al., 1995). In the Mediterranean Sea, the altimetry-derived geostrophic velocity is generally smaller than the drifter observations (Poulain et al., 2012); the error increases with the geostrophic velocity with an error of 7-17% at the velocity of 1.5 m s$^{-1}$ (Kubryakov and Stanichny, 2013).

The satellite ADT is linearly interpolated onto the 1/3° OSCAR grid (Figure 4). The magnitude of the geostrophic current is generally comparable to that of the total current, although their directions are not perfectly aligned. Large geostrophic velocity is observed along the western boundary of the GoT and at the southeastern entrance. The geostrophic velocity is weak in the GoT interior. The root-mean-square (rms) difference between the total and the geostrophic current (i.e. the estimated ageostrophic current) ranges from 0.04 to 0.11 m s$^{-1}$ with the largest difference observed along the western boundary of the GoT; however, the rms difference is reasonably proportional to the speed of the total current there. A complex correlation between the total and geostrophic current is calculated to determine the correlation and phase relationship between the two velocity fields (Figure 6a, b; Eq. 7). Over the entire basin, the rms correlation coefficient is 0.70 (Figure 6a). The correlation between the total and geostrophic current is higher along the southern boundary of the observing domain, at 8° N. Along the northeastern boundary, geostrophic current only explains a small fraction of the total current variance (14 − 34%). The rms correlation coefficient over the interior is 0.71 and that over the western boundary region is 0.67. The strong correlation indicates the dominance of geostrophic circulation over the GoT. Phase relationship shows the direction that the geostrophic current has to rotate to align with the direction of the total current where positive denotes counterclockwise rotation. A negative phase relationship is found roughly between 10.5° and 12° N while a positive relationship is found to the south of 10.5° N (Figure 6b). With the presence of anticyclonic circulation centered at 10.5°-11° N during the southwest monsoon and fall monsoon transition, the phase relationship requires a southeastward ageostrophic flow. In contrast, the phase relationship implies a northwestward ageostrophic current during the northeast monsoon and spring monsoon transition when cyclonic circulation dominates the GoT interior. As the prevailing monsoon wind is southwesterly during the southwest monsoon and northeasterly during the northeast monsoon, the resulting wind-driven Ekman current aligns with the direction of the ageostrophic flow in the respective seasons. Therefore, the Ekman current is calculated from the CCMPv2 wind (Eq. 3 and 4) to compare with the ageostrophic current.

Wind-driven Ekman current averaged in the upper 30 m of the water column (or to the seafloor where the water column is shallower than 30 m) has a distinct seasonal cycle being the strongest during the northeast monsoon and the weakest during the

spring monsoon transitions (Figure 4b, d, f, h). During the northeast monsoon, the speed of the Ekman current exceeds 0.06 m s$^{-1}$ almost everywhere except between 10° - 12° N along the eastern coast. The weak Ekman current is likely due to the presence of the Cardamom Mountains between 10° - 13° N at the coast of Thailand and Cambodia that blocks the northeasterlies (Li et al., 2014). The strongest current is at the southeastern entrance with the speed of 0.1 m s$^{-1}$ transporting water into the GoT (Figure 4h). The Ekman current at the southeastern entrance still transports water into the GoT during the spring monsoon transition, although the speed decreases due to the weakening of the monsoon wind (Figure 4b). During the southwest monsoon and fall monsoon transition, the Ekman current is quite uniform over the entire GoT; the flow is southeastward producing an outflow into the SCS (Figure 4d, f). The magnitude of the Ekman current is similar to the ageostrophic current in the GoT interior and the southeastern entrance, but smaller along the boundaries.

Complex correlation between the ageostrophic component and wind-driven Ekman current is calculated to examine the contribution of wind-driven current on the ageostrophic circulation (Figure 6c). Stronger correlation is found over the southern part of the domain with the largest correlation coefficient of 0.54 reflecting that up to 29% of the ageostrophic circulation is wind-driven. Over the western boundary region, the correlation coefficient between the wind-driven current and the ageostrophic current is 0.40. A similar value of correlation coefficient is observed over the interior of the GoT ($R = 0.41$). Phase relationship indicates the direction that the Ekman current has to rotate to align with the direction of the ageostrophic current; it is small overall with most values between $-\frac{\pi}{4}$ and $\frac{\pi}{4}$ (Figure 6d). When ageostrophic current is entirely driven by wind stress, the phase relationship is zero. Thus, the non-zero phase relationships hint at the importance of forcings other than wind stress on ageostrophic current, e.g. counterflow produced by the bottom friction (the bottom Ekman layer), etc. The negative phase is only found in a narrow band along the northwestern boundary suggesting the direction of the wind-driven current that is to the left of ageostrophic flow. The negative phase is clearly evident in June, September, and December, while a positive phase is observed during the spring monsoon transition (March) (Figure 4b, d, f, h).

Although the spatial resolution of the OSCAR products is too coarse to capture the complex circulation over such small and shallow basin as the uGoT, previous studies have described the general circulation (Yanagi et al., 2001; Buranapratheprat et al., 2006, 2008; Saramul and Ezer, 2014) and thus speculation on the interaction between the uGoT and the rest of the GoT is discussed below. During the southwest monsoon, a southward flow is present along the western boundary to the south of ∼12.5° N (Figure 4c) and the uGoT circulation is anticyclonic; thus, the flow along the GoT western boundary is likely not continuous into the uGoT. Instead, the northward flow along the western boundary of the uGoT is likely supplied by the western flank of the anticyclonic circulation in the GoT interior transporting high salinity (Buranapratheprat et al., 2002) and low nutrient (Buranapratheprat et al., 2009) water into the uGoT (Figure 1a). The southward current along the uGoT eastern boundary is confluent with the eastward/ northeastward flow to the southeast of the uGoT. During the northeast monsoon, a cyclonic circulation is present over the uGoT. The southward flow along the uGoT western boundary potentially continues southward joining the western flank of the cyclonic circulation in the GoT interior (Figure 1a, 4g). Along the uGoT eastern boundary, the northward current is supplied by water from lower latitudes with relatively high salinity and low-nutrient (Buranapratheprat

et al., 2002, 2009). The water mass is likely derived from the westward/ northwestward flow between 11° and 12° N that bifurcates to have one flank travels northward along the uGoT eastern boundary and the other continues westward serving as the northern flank of the GoT interior cyclonic eddy. As studies on circulation patterns during the monsoon transition are still quite scarce, speculation regarding the circulation between the uGoT and the rest of the GoT is omitted.

### 4.1.3 The role of wind stress curl on the sea surface height

The impact of the wind stress curl on sea surface height is examined through linear regression using daily measurements at the original spatial resolution of 1/4°. Both the basin-averaged wind stress curl and ADT have distinct seasonal cycles. High (low) ADT is observed during the northeast (southwest) monsoon over the entire basin, while the wind stress curl exhibits the opposite pattern (Figure 7a). In addition, the intraseasonal signals of the basin-averaged ADT covary quite well with the basin-averaged wind stress curl. The anticorrelation yields a significant negative correlation of -0.84 indicating the dominance of wind stress curl on the ADT through local Ekman pumping; positive (negative) wind stress curl induces upward (downward) flow in the water column and depresses (raises) the ADT. In addition, the correlation suggests no time lag between the wind curl and the ADT reflecting the instantaneous adjustment of the ADT as being forced by the local wind stress curl. As Zhou et al. (2012) suggest a delayed response of the ADT to wind stress curl over the SCS (decaying timescale of ∼40 days), the result demonstrates the different wind-associated dynamic that underlies the GoT compared to the rest of the SCS.

As most energy of the GoT circulation locates along the western boundary of the GoT (Figure 2, 3, 4), the influence of wind stress curl on current variability at 9.6° N, 99.9° E representing high ADT variance to the south of the uGoT (purple cross in Figure 2b) and at 12.9° N, 100.1° E representing high variance along the western GoT (green cross in Figure 2b) are further investigated. Since an effect of winds on the ocean circulation is not necessarily local nor applied over a large scale (e.g., Meyers, 1996; Giddings and MacCready, 2017), the relationship between ADT at the selected locations and wind stress curl over the entire GoT is examined to identify the location of wind stress curl that influences the ADT. The ADT with high variance to the south of the uGoT correlates well with the nearby wind stress curl with a correlation coefficient of -0.80. The high correlation suggests a local response of ADT and thus the geostrophic current to the wind stress curl, although much of the correlation is attributed to the dominant seasonal cycle observed in both variables (Figure 2b, 7c). However, high-variance ADT at the western boundary is correlated with wind stress curl ∼280 km to its north (Figure 2b, 7b). Wind stress curl to the south of the uGoT (12.1° N, 100.1° E) explains 80% of the ADT variance at the central part of the GoT western boundary (9.6° N, 99.9° E) as it also captures the intraseasonal fluctuation of the ADT. The mechanism associated with the high correlation between ADT at the western boundary and remote wind stress curl is still unclear and beyond the scope of this study. Still, the result suggests the importance of coastal trapped Kelvin waves which travel equatorward along the western boundary of the basin (Wang, 2002). Coastal trapped Kelvin waves are also commonly found in regions with shallow and complex bathymetry, e.g., the Indonesian Archipelago (Sprintall et al., 2000; Delman et al., 2018), the SCS, and the East China Sea (Wang et al., 2003; Yin et al., 2014; Liu et al., 2011). At the GoT interior, wind stress curl does not exhibit a seasonal cycle, and thus, the local wind stress curl does not locally influence the ADT there.

### 4.1.4 Interaction with the South China Sea

Since the GoT connects to the SCS, variability of the SCS circulation would provide a better understanding of the GoT circulation and the origin of water masses transported into the basin. In the southern part of the SCS, the circulation is highly influenced by the monsoon winds (e.g. Hu et al., 2000; Gan et al., 2006). During the northeast monsoon when the inflow from the SCS to the GoT is observed (Figure 4g), a strong southwestward flow is present off the eastern coast of Vietnam; the current partly turns northwestward transporting water into the GoT (Hu et al., 2000; Gan et al., 2006; Liu et al., 2008). Thus, the GoT is largely replenished by water from the northern SCS which is highly influenced by the Kuroshio intrusion (Chao et al., 1996; Jilan, 2004; Wang et al., 2006; Centurioni et al., 2009). As the Kuroshio intrusion path can be quite variable, the northern SCS circulation varies depending on the intrusion path (Hu et al., 2000; Caruso et al., 2006; Nan et al., 2015). The variable SCS circulation potentially contributes to the variability of the GoT circulation, particularly at the entrance. During the southwest monsoon, a northeastward flow is present to the south of the GoT and the current off the Vietnam coast reverses to flow northwestward (Hu et al., 2000; Gan et al., 2006; Liu et al., 2008). The observed GoT outflows during the southwest monsoon and fall monsoon transition likely join the northeastward flow transporting freshwater from river runoffs (Aschariyaphotha and Wongwises, 2012) into the SCS.

### 4.2 Interannual variability of the circulation in the Gulf of Thailand

The influences of ENSO and IOD are examined to understand the interannual variability of the GoT circulation. Low-frequency OSCAR velocity is calculated by removing the seasonal cycle, taken as a linear combination of the annual and semiannual harmonics that best fit the 6-year observations, and signals with periods shorter than 90 days. A 90-day lowpass filter is also applied to Niño3.4 and DMI (see Datasets section). Complex correlation between the low-frequency indices, which have a phase of zero ($\pm\pi$) for a positive (negative) value, and the low-frequency currents is calculated over the entire GoT (Figure 8a, b). The correlation shows fascinating patterns revealing that the ENSO conditions strongly influence the circulation over the central and the eastern parts of the GoT, while IOD conditions influence the current along the western boundary of the GoT and the southern boundary of the observing domain.

Along the western boundary of the GoT, strong correlation between the low-frequency current and low-frequency DMI is found with a phase relationship of $\frac{\pi}{2}$ indicating northward (southward) current anomaly during a positive (negative) IOD condition (Figure 8b). The low-frequency meridional current averaged in 9.0° - 11.5° N, 99.5° - 100.2° E region is used to represent the current along the western boundary; low-frequency DMI explains 45% of the variance of the alongshore meridional flow (Figure 9b). During the southwest monsoon and fall monsoon transition in 2016 when a negative IOD event occurs, the southward western boundary current (Figure 4) intensifies. In contrast, the seasonal southward current significantly weakens during the southwest monsoon and fall monsoon transition of 2019 when a positive IOD occurs. In addition, DMI correlates with the OSCAR current along the southern boundary of the domain. With the phase relationship of approximately $\pm\pi$, the current is westward (eastward) along ~8° N during a positive (negative) IOD event. The result suggests that IOD events do not only

affect the current along the western boundary but they also impact the continuous current from the southeastern entrance to the western boundary of the GoT (Figure 3-5). The results are contrasting with the finding by Higuchi et al. (2020) that suggests an

355 anomalous outflow during the southwest monsoon season of a positive IOD event. To understand the dynamics associated with the low-frequency variability of the GoT circulation, correlations between low-frequency DMI and selected forcings, which are ADT, zonal wind stress, and wind stress curl, are calculated (Figure 8c, e, g). Similarly, those between Niño3.4 and the selected forcing are also computed (Figure 8d, f, h). Note that the meridional wind stress is also considered; however, its correlation with either of the indices does not exhibit a distinct variation pattern over the GoT. The low-frequency variability of the current

along the western boundary during IOD events is likely associated with local zonal wind stress (Figure 8f). Low-frequency component of the zonal wind stress shows strong correlation with the DMI along the GoT western boundary with correlation coefficients as large as -0.75 and along the southern boundary of the domain with correlation coefficients as large as -0.58; the correlation pattern is similar to that between the OSCAR current and DMI (Figure 8b). The negative correlation suggests the westward (eastward) wind stress anomaly during a positive (negative) IOD event yielding a northwestward (southeastward)

surface Ekman current anomaly along the western boundary, consistent with the low-frequency OSCAR current. Although the low-frequency wind stress curl and ADT also suggest northward flow anomaly along the western boundary, the influence is roughly the same along the entire western boundary as well as the eastern boundary (Figure 8d, f). Therefore, the low-frequency variability of the current along the GoT western boundary is likely not associated with the ADT and wind stress curl variability during IOD conditions. Correlation between low-frequency Niño3.4 and low-frequency meridional current is small ($R = 0.14$)

but significant with 95% confidence suggesting a tendency of a northward (southward) current anomaly during an El Niño (La Niña) event; however, the current anomaly is not clearly apparent during the 2015/2016 El Niño and the 2017/2018 La Niña (Figure 9b).

In the GoT interior, a region of strong correlation between Niño3.4 and OSCAR current is found at ∼10.0° - 11.0° N, 100.5°

375 - 103.0° E (the northern dashed box in Figure 8a), with the out-of-phase relationship that indicates westward current anomaly during an El Niño event and eastward current anomaly during a La Niña event. Considering low-frequency zonal current averaged over the region, Niño3.4 explains 34% of the low-frequency variance of the zonal flow (Figure 9c). With a 75-day lag of the area-averaged current, the correlation improves to -0.71; half of the variance in low-frequency zonal current at this region is associated with ENSO condition. Along the southern boundary of the domain (∼8° - 9° N, 101.5° - 104.0° E), the correlation

is also strong and significant with 95% confidence. However, the relationship is in-phase indicating an eastward (westward) current anomaly during an El Niño (a La Niña) event. This results in a weak GoT inflow at the southeastern entrance during an El Niño event consistent with weak circulation found in the SCS (Chao et al., 1996; Wang et al., 2006). The low-frequency Niño3.4 explains 14% of variance in the area-averaged low-frequency zonal current with no lag (Figure 9c). This opposing pattern produces a cyclonic current anomaly in the GoT interior during an El Niño and an anticyclonic anomaly during a La

Niña with its location farther south than the seasonal cyclonic and anticyclonic circulation (Figure 3, 4a, c). The low-frequency cyclonic (anticyclonic) circulation during an El Niño event (a La Niña) colocates with anomalously low (high) ADT suggesting the dominance of geostrophic response at the GoT interior to ENSO variability (Figure 8c). The mechanism setting up the

low-frequency ADT variability is unclear; it cannot be explained by the local wind stress and wind stress curl (Figure 8e, g) but might relate to the winter warm pool located at the eastern boundary of the GoT (Li et al., 2014). Still, the influence of low-frequency wind stress curl on the ADT through coastal trapped Kelvin waves cannot be eliminated. The correlation map suggests a negative (positive) wind stress curl at the GoT eastern boundary between 8.5° and 11.5° N during an El Niño (La Niña) yielding a positive (negative) ADT anomaly that could propagate northward in the form of coastal trapped Kelvin waves. During the 2015/2016 northeast monsoon, the cyclonic circulation centered at 9.7° N, 101.7° E is clearly apparent during the northeast monsoon and into the spring monsoon transition (Figure 9c), while the seasonal cyclonic circulation centered at 10.5° N, 101.5° E is missing (Figure 3-4). Therefore, the seasonal cyclonic circulation locates farther south during the El Niño event. Shortly after the peak of the 2017/2018 La Niña, an anticyclonic anomaly centered at the same location (9.7° N, 101.7° E) develops (Figure 9c). The anticyclonic eddy anomaly in the GoT interior becomes more pronounced during the 2018 spring monsoon transition when the La Niña decays. The low-frequency DMI is significantly correlated with the low-frequency zonal current in the interior. However, it explains less than 5% of the low-frequency zonal current variance, and the general circulation pattern away from the western boundary and the uGoT do not show a significant deviation from the seasonal current during the negative IOD event in 2016 and positive IOD event in 2019 (Figure 8b, 9c).

The ADT, zonal wind stress, and wind stress curl are also examined over the uGoT region to understand how these forcings, which potentially influence the uGoT interannual circulation, vary during the ENSO and IOD events (Figure 8). Positive correlation is found between the ADT over the uGoT and Niño3.4 indicating a tendency of an anomalously high (low) sea level, particularly along the eastern boundary during an El Niño (La Niña) event (Figure 8c). The pattern is consistent with that produced by the wind stress curl (Figure 8g). Therefore, the geostrophic current is likely anomalously northward (southward) along the western boundary of the uGoT during an El Niño (La Niña) event. In addition, a positive correlation between the zonal wind stress and Niño3.4 is present reflecting anomalously southward (northward) wind-driven Ekman current during an El Niño (La Niña). Correlations between the selected forcings and DMI are generally lower than those with the Niño3.4 (Figure 8c-h). The correlation between DMI and ADT is weak over the northwestern part of the uGoT; higher positive correlation is observed to the south of the uGoT and along the eastern boundary. As a result, anomalously high (low) ADT is likely present to the south of the uGoT and the eastern boundary during a positive (negative) IOD event producing an eastward (westward) geostrophic current to the south of the uGoT and a northward (wouthward) geostrophic current along the eastern boundary (Figure 8d). A positive correlation between DMI and wind stress curl is also found over the uGoT potentially contributing to a lower increase in the sea level compared to that in the rest of the GoT where negative correlation is present (Figure 8f). IOD events are overall not significantly correlated to the zonal wind stress over the uGoT (Figure 8f).

## 5  Conclusions

This study exploits the synergy of the available remotely-sensed observations to understand variability of the GoT circulation that reveals different responses to the different climate modes. The interannual current along the western boundary is more sensitive to IOD conditions, while that in the GoT interior is more sensitive to ENSO conditions (Figure 8, 9). At the seasonal timescale, the observations reveal spontaneous adjustment of the basin-averaged ADT following the basin-averaged wind stress curl signal that is different from the rest of the SCS (Zhou et al., 2012). Still, the associated mechanisms vary over different

parts of the GoT (Figure 7). For example, the ADT at the southern part of the uGoT highly correlates with the local wind stress curl reflecting the influence of the local Ekman pumping, while the ADT along the western boundary is highly related to the wind stress curl to its north suggesting the influence of coastal trapped Kelvin waves on modifying the sea level along the western boundary. Approximately half of the surface current variability is geostrophic (Figure 4, 6) set up by ADT variability similar to circulation in the SCS (Gan et al., 2006). The ageostrophic current is significantly explained by the Ekman current,

although the portion being explained varies spatially. The Ekman current accounts for a larger percentage of the ageostrophic flow along the western boundary and the GoT interior, particularly to the south of ∼10° N, compared to region to the north of this latitude (Figure 4, 6).

The OSCAR surface current product also demonstrates the seasonal reversing circulation pattern at the surface, explaining

28% of the total current variance over the 2014 - 2019 period, following the monsoon wind reversal (Figure 3a-c). The seasonal pattern confirms the anticyclonic circulation in the GoT interior with an outflow at the southeastern entrance during the southwest monsoon (Figure 4) consistent with findings from Wyrtki (1961) and Aschariyaphotha et al. (2008). A cyclonic circulation along the western boundary of the GoT as suggested by a numerical simulation (Yanagi and Takao, 1998) is present but narrow, confined to the west of ∼100.5° E (Figure 1a). Although the anticyclonic geostrophic circulation at the rim of the GoT (So-

jisuporn et al., 2010) is not distinct, the southward geostrophic flow along the western boundary is strong. During the northeast monsoon, OSCAR product shows strong northward flow along the western boundary and an inflow at the southern entrance consistent with observational (Sojisuporn et al., 2010) and numerical studies (Yanagi and Takao, 1998; Aschariyaphotha et al., 2008). The 6-year averaged surface velocity displays the dominance of cyclonic circulation in the GoT interior (Figure 4), in agreement with that observed during the NAGA expedition (Wyrtki, 1961).

In addition, the finding reveals the importance of the GoT circulation during the monsoon transitions; there are strong seasonal flows along the western boundary and at the southeastern entrance suggesting water exchange with the SCS. During the spring monsoon transition, a strong northward flow along the western boundary superimposes on the circulation pattern during the northeast monsoon (Figure 3-4). In contrast, a southward flow is present along the western boundary and strong

southeastward current is observed at the southeastern entrance during the fall monsoon transition. As the western boundary current is connected to that at the southeastern entrance (Figure 5), the results highlight the connection between circulation in the GoT and the SCS that distinctly occurs during the monsoon transitions. Moreover, variability of the circulation during the

transition seasons could largely impact the properties of water in the GoT.

Although this study offers insightful details of the GoT circulation, particularly to the south of 12.5° N, there are still forcings that are not considered, for example, tidal currents and planetary waves. Previous studies have indicated the importance of tidal pattern over the GoT that is different from one region to another (Yanagi and Takao, 1998) and is heavily dependent on tidal pattern in the SCS (Cui et al., 2019). The effect of coastal trapped Kelvin waves on the coastal circulation which is also observed in the nearby regions (Sprintall et al., 2000; Wang et al., 2003; Yin et al., 2014; Liu et al., 2011; Delman et al., 2018), cannot

be concluded despite the observed influence of remote wind stress curl on the western boundary current. Thus, examination of these factors could further improve the understanding of dynamics associated with the GoT circulation.

*Data availability.* The OSCAR third degree Version 1 data are provided by Physical Oceanography Distributed Active Archive Center. CCMP Version-2.0 vector wind analyses available at www.remss.com are produced by Remote Sensing Systems. This study has been conducted using E.U. Copernicus Marine Service Information (http://marine.copernicus.eu/services-portfolio/access-to-products/?option=com_

csw&view=details&product_id=SEALEVEL_GLO_PHY_L4_REP_OBSERVATIONS_008_047). Niño 3.4 and DMI are available at https://www.cpc.ncep.noaa.gov/data/indices/wksst8110.for and https://stateoftheocean.osmc.noaa.gov/sur/ind/dmi.php, respectively.

*Author contributions.* The author confirms contribution to the paper as follows: data collection: A. Anutaliya; data analysis: A. Anutaliya; discussion of results: A. Anutaliya.

*Competing interests.* The author declares no relevant competing interests.

*Acknowledgements.* This work has been supported by the 2021 Research Grant of Burapha University under grant 001/2564 to Arachaporn Anutaliya.

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

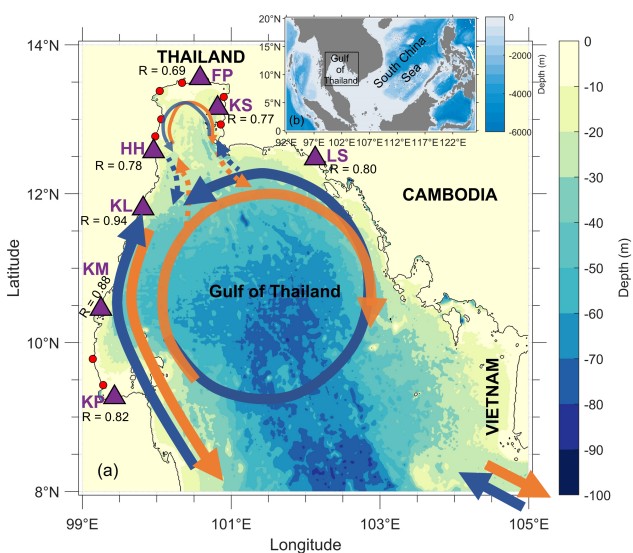

**Figure 1.** Map of the Gulf of Thailand (GoT). (a) Schematic of the surface current to the south of ∼12.5° N suggested by this study (thick line) and that to the north of ∼12.5° N derived from previous studies (e.g., Yanagi et al., 2001; Buranaprateprat et al., 2006, 2008; Saramul and Ezer, 2014, thin line) during the southwest monsoon (orange) and the northeast monsoon (blue). Dashed line denotes suggested circulation between the uGoT and the rest of the GoT. Triangles indicate the locations of tide gauges at KP (Ko Prap), KM (Ko Mattaphon), KL (Ko Lak), HH (Huahin), FP (Fort Phrachula Chomklao), KS (Ko Sichang), and LS (Laem Sing) with values of correlation coefficients between measurements from the tide gauges and satellite altimetry. (b) Shows the location of the GoT. Color contours in (a) and (b) represent bathymetry and the black contour in (a) represents the zero-depth level.

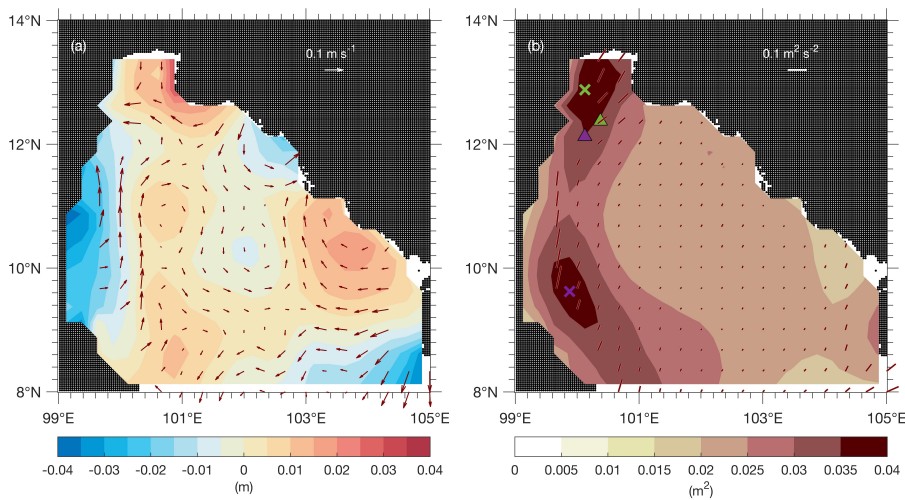

**Figure 2.** (a) The 2014-2019 (a) mean and (b) variance of OSCAR current (maroon quiver/ line) and ADT (color contour) over the GoT. The zonal (meridional) component of the maroon lines in (b) indicates variance of the zonal (meridional) OSCAR current. Crosses in (b) mark locations with high ADT variance in the uGoT (green) and at the GoT western boundary (purple); the triangles with respective colors mark the locations of wind stress curls that correlate the best with the ADT shown in Figure 7.

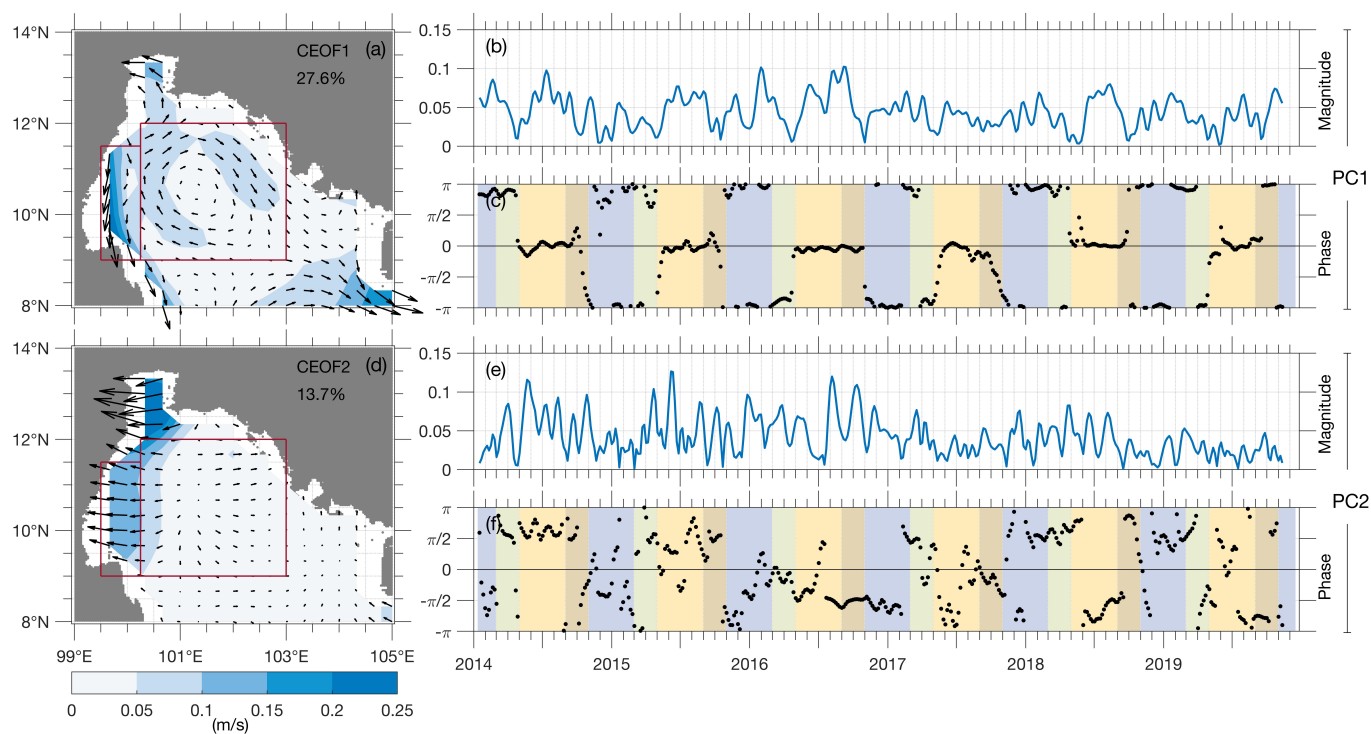

**Figure 3.** The (a) first and (d) second dominant complex empirical orthogonal function (CEOF) modes of the OSCAR current with (b), (e) the corresponding magnitudes and (c), (f) phases of the principal components (PC). Percentage of the current variance explained by each mode is shown on the upper right corner of (a) and (d). Maroon boxes in (a) and (d) designate the western boundary region (9.0° - 11.5° N, 99.5° - 100.2° E) and the GoT interior region (9.0° - 12.0° N, 100.2° - 103.0° E). Background shading in (c) and (f) denotes different seasons: northeast monsoon (blue), spring monsoon transition (green), southwest monsoon (yellow), and fall monsoon transition (brown).

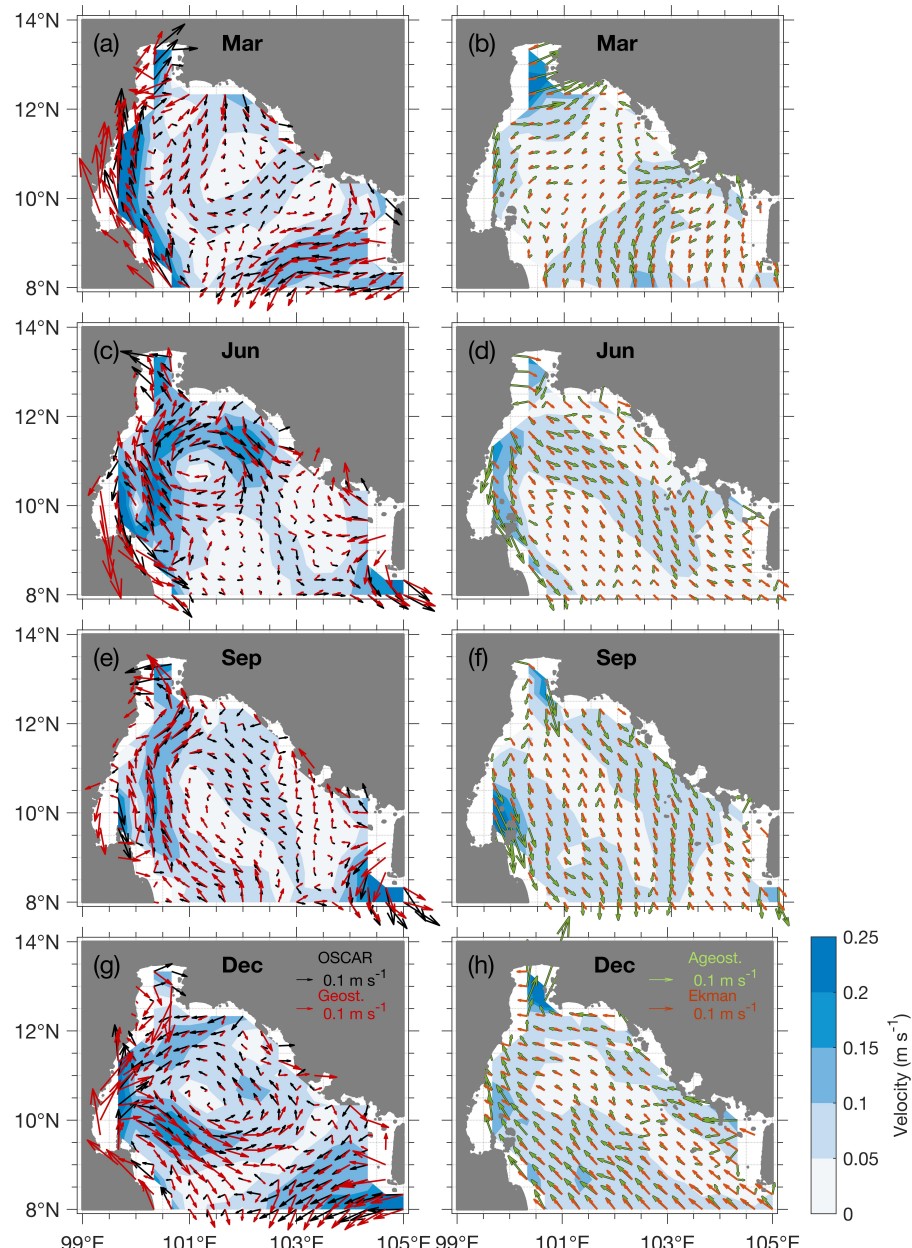

**Figure 4.** The 2014 - 2019 monthly mean current over the GoT from OSCAR product (black quiver; left column) and altimetry-based geostrophic current (red quiver; left column) for (a) January, (c) June, (e) September, and (g) December; color contour indicates magnitude of the OSCAR current. The monthly mean ageostrophic current (green; right column) and wind-driven Ekman current (brown; right column) for (b) January, (d) June, (f) September, and (h) December; color contour indicates magnitude of ageostrophic current. The color scale is presented on the lower right corner and the scale for the current is shown in (g) and (h).

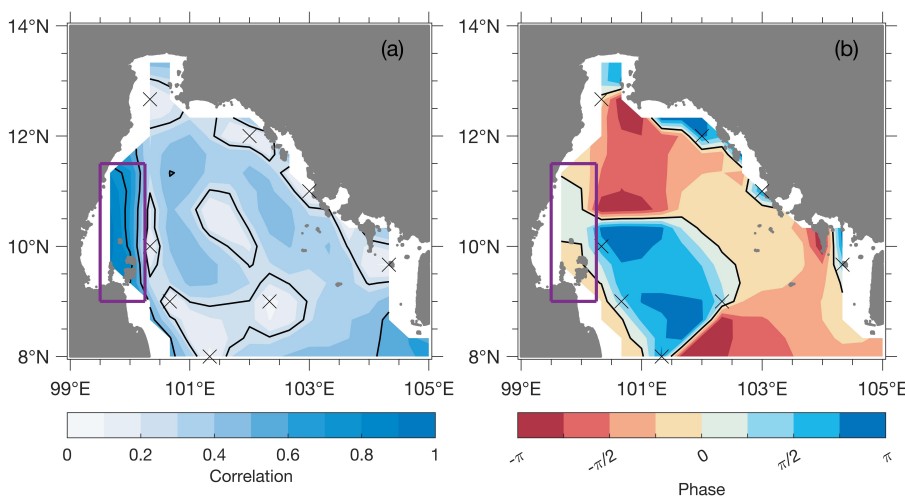

**Figure 5.** (a) Complex correlation between OSCAR current along the western boundary (purple box) and OSCAR current over the entire GoT and (b) the corresponding phase. Black contour in (a) is plotted every 0.25 and that in (b) is plotted at zero and $\pm\pi$. Crosses indicate the regions where the correlation is not significant with 95% confidence.

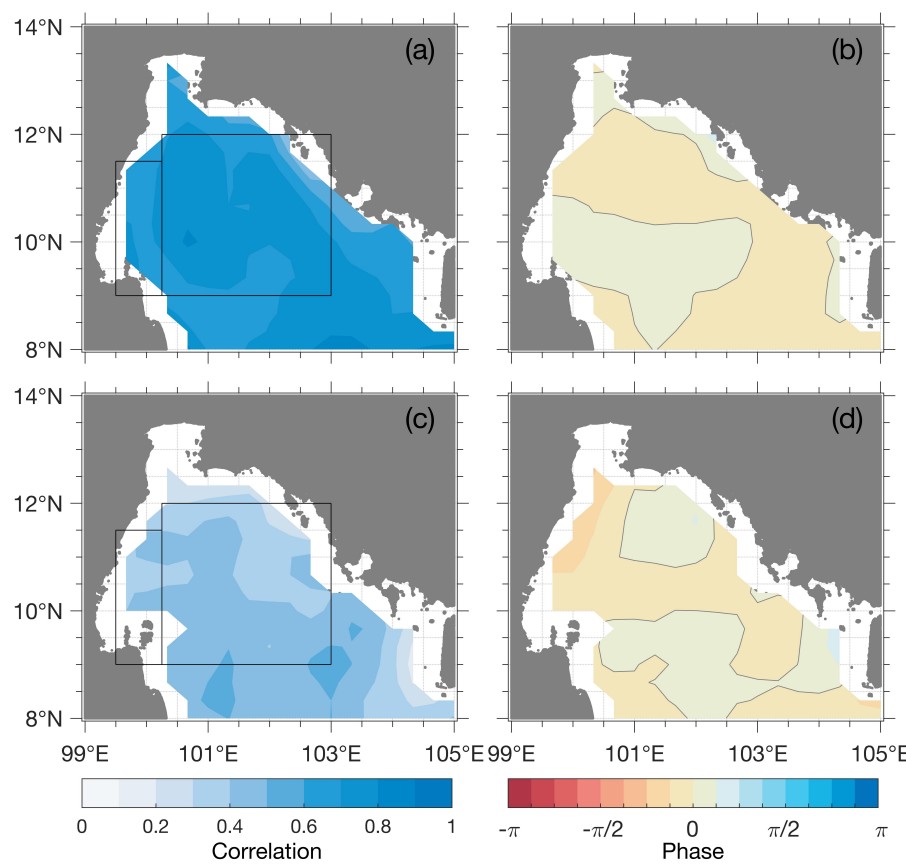

**Figure 6.** (a) Complex correlation map between the OSCAR current and altimetry-derived geostrophic current and (b) the phase. (c) Complex correlation map between the ageostrophic current and the Ekman current and (d) the phase. Boxes in (a) and (c) designate the western boundary region (9.0° - 11.5° N, 99.5° - 100.2° E) and the Gulf of Thailand interior region (9.0° - 12.0° N, 100.2° - 103.0° E).

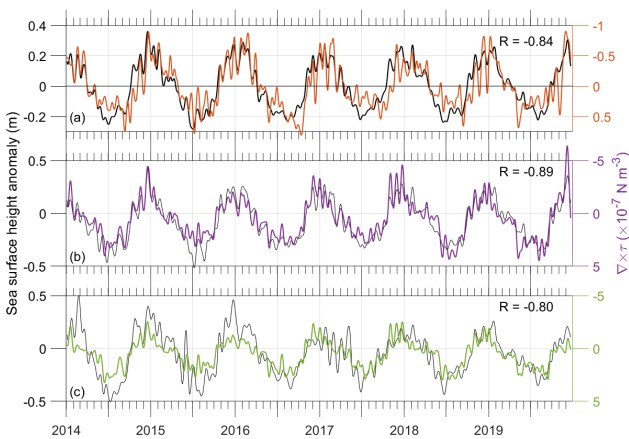

**Figure 7.** Comparison between sea surface height anomaly (black) and wind stress curl (colors): (a) both sea surface height anomaly and wind stress curl (orange) averaged over the entire GoT, (b) sea surface height anomaly at the GoT western boundary shown as purple cross in Figure 2b and wind stress curl (purple) to the south of the uGoT shown as purple triangle in Figure 2b, and (c) both sea surface height anomaly and wind stress curl (green) to the south of the uGoT shown as green cross and triangle in Figure 2b. Correlation coefficient between each comparison is shown on the upper right corner of each subplot. Note the reversed y-axis for wind stress curl.

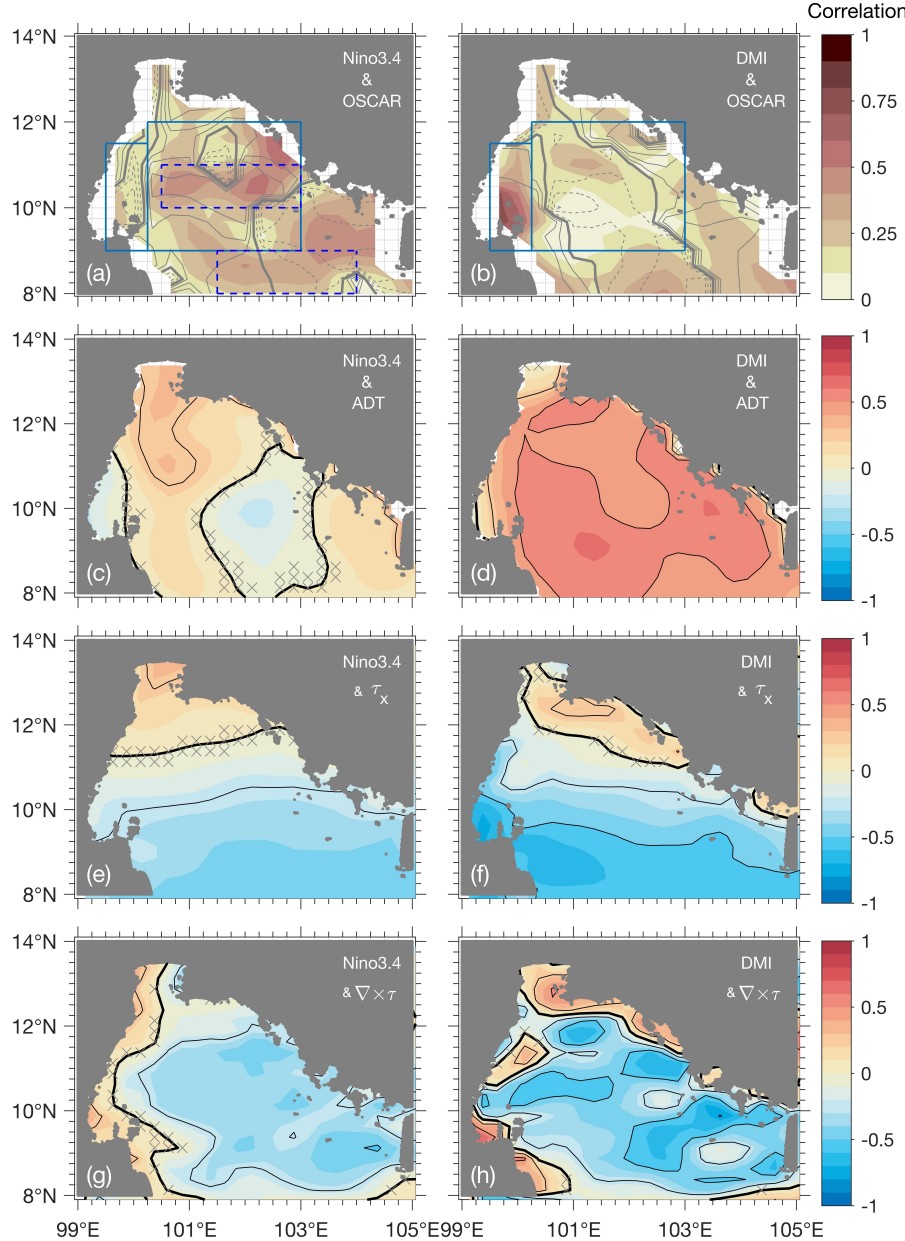

**Figure 8.** Complex correlation was calculated between OSCAR current and the climate modes: Niño 3.4 ((a); left column) and Dipole Mode Index ((b); right column) with gray contour showing the phase associated with the complex correlation map. Solid (dashed) gray line is plotted every $\frac{\pi}{4}$ showing the positive (negative) phase relationship; thick gray line designates the zero contour. Correlation maps between the climate modes and the selected forcings: (c), (d) sea surface height, (e), (f) zonal wind stress, (g), (h) wind stress curl. The blue boxes in (a) and (b) designate the western boundary region (9.0° - 11.5° N, 99.5° - 100.2° E) and the GoT interior (9.0° - 12.0° N, 100.2° - 103.0° E). Dashed blue boxes in (a) are areas with large low-frequency variability presented in Figure 9c. Thick black contour in (c) – (h) is plotted at zero and thin black contour is plotted every 0.25. Crosses in (c) - (h) mark the regions where the correlation is not significant with 95% confidence.

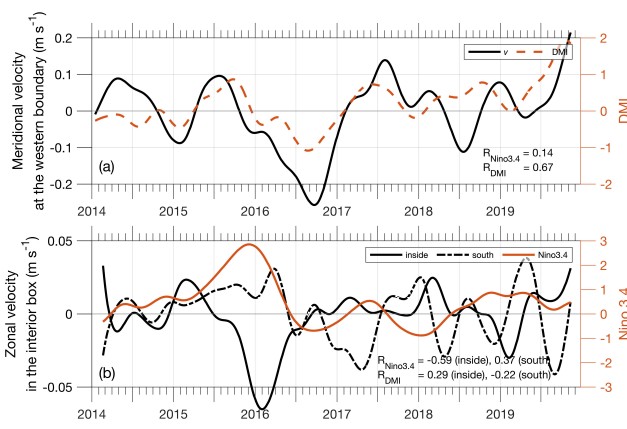

**Figure 9.** Comparison between low-frequency current and the climate modes: (a) meridional velocity averaged along the western boundary (solid black) and the Dipole Mode Index (dashed orange), (b) mean zonal velocity in the middle (solid black) and to the south (dashed black) of the GoT interior and Niño 3.4 index (solid orange). Correlation coefficients between the current and both climate modes are shown on the lower right corner of each subplot. The correlation coefficients between the climate modes and the mean zonal velocity in the middle (to the south) of interior box are shown as the first (second) values.