# Peer review of "Surface circulation in the Gulf of Thailand from remotely sensed observations: seasonal and interannual timescales"

_EGUsphere, 2022_

## Author Response (AR1)

**Surface circulation in the Gulf of Thailand from remotely sensed observations: seasonal and interannual timescales**

Arachaporn Anutaliya

The author comment is presented in the following sequence: (1) itemized comments from the editor in black, (2) author's response in red, (3) the revised text is in red italic.

Thank to the reviewers for your helpful comments.

Anonymous Reviewer #1:

The manuscript describes the seasonal and interannual surface circulation variability in the Gulf of Thailand (GoT) through the analysis of remote sensed data (i.e., SST, SSH, etc.). Complex EOF and complex correlation and other spatio-temporal analysis were taken into consideration. Since this is, I think the first paper that tries to explain the dynamics of surface circulation in the GoT using observation (remote sensing) data, more detail explanation is necessary. I think not only oceanographers want to know the surface circulation in the GoT, but also other scientists (marine biologists, chemical oceanographers, fishery scientists, etc.). So, my comments will be pointed out in the following.

Major comments:

1. Line 16: author defined the GoT domain (6-14 N and 98-106 E), but when author begins to analyze the data, why author defined the analyzed area as 8-14 N (Line 73) not 6-14 N. During SW monsoon, northward flow from East Malaysian Peninsula could also play significant role for the circulation in the GoT as is showed the inflow from South China Sea (SCS) during NE monsoon. Please clarify this.

    1. Thank the reviewer for pointing it out. The GoT domain is redefined to be between 8-14° N (L15).

    2. Discussion relating to the influence of the South China Sea and that along the East Malaysian Peninsular is added.

    *(L16) On seasonal timescale, the Asian monsoon winds prevail producing a wet season over southeast Asia during the southwest monsoon (May to August) and a dry season during the northeast monsoon (November to February). Although the GoT circulation also heavily dependent on inflows from the SCS, e.g., along the eastern coast of Malaysia and around the southern coast of Vietnam, these currents are mainly driven by the monsoon winds (Wyrtki, 1961; Akhir, 2012).*

2. Line 78-80: when author mentioned that the largest difference between OSCAR and HF Radar is found at UGoT where there are only 6 OSCAR data points. Does author try to conclude that largest difference found at UGoT is because of there are only 6 OSCAR data points? What about other areas in the GoT?

    1. The large difference between the OSCAR data and the coastal-radar found at the uGoT is essentially due to the spatial resolution of the OSCAR data that is too coarse for the area. As a result, there are only a few measurements in the uGoT regions (only 6). The author understands that the original text is quite misleading and has revised to better explain the point. The available OSCAR data in each defined region is presented in Figure 3; there are 14 data points in the western boundary box and 72 data points in the interior box.

    *(L87) The difference between OSCAR velocity and high-frequency coastal-*

*radar velocity is the largest in the uGoT; as the region is quite small and shallow (Figure 1), the spatial resolution provided by the OSCAR products might not be sufficient to resolve the circulation there.*

3. Line 83-87: Please clarify sea surface level (is it monthly data? And what is the source of ADT data?). Do you think 3 stations situated in the UGoT and western GoT are enough to validate satellite-derived ADT? What about the stations in the eastern GoT? I think there are available data from local authorities that you can request. When author mentioned western boundary, does it include the western side of UGoT (no data for OSCAR).

    1. The ADT is the gridded all-satellite merged product provided at daily resolution. See Line 97.

    2. Thanks to the reviewer for the suggestion to also compare the ADT data to the available tide gauge along the eastern boundary. Further ADT validation against four additional tide gauge stations is carried out with the two out of four stations located along the eastern side of the GoT (Laem Sing and Ko Sichang). The calculated correlation coefficients between the ADT and tide gauge data are presented in Figure 1.

    *(L98) As the satellite altimetry used here is in the coastal region (Figure 1), sea surface level data from 7 tide gauge stations in the GoT, which are Fort Phrachula Chomklao (FP), Ko Lak (KL), Ko Mattaphon (KM), Huahin (HH), Ko Prap (KP), Laem Sing (LS), and Ko Sichang (KS) (Holgate et al., 2013; PSMSL, 2019), are used to validate the satellite-derived ADT. The comparisons show high correlation between the fluctuation of satellite ADT and the tide gauge sea level over the 2014-2019 period with correlation coefficients ranging from 0.69 to 0.94 as shown in Figure 1.*

    *3.* The author does not locate 'western boundary' in the specified line. The author is more careful to specify which part of the western boundary the author is referring to throughout the manuscript.

4. Line 99: How to estimate A?

    *1.* The estimation of A is added.

    *(L115) …   $A=8\times10^{-5}|U|^{2.2}$).*

5. Line 111-113: Why high variance of ADT indicating the influence of geostatic flow and low variance indicating ageostrophic flow? Could you give a reference for that statement?

    1. The geostrophic current can be directly estimated from the satellite-derived ADT. The relationship is added to the datasets section and the reference to these equations is inserted in the text. Therefore, the change (hence variance) of ADT directly associates with the geostrophic current. On the contrary, geostrophic current is suggest to not be important when the ADT barely fluctuates (low variance), and thus the ageostrophic component could play an important role.

    *(L92) The gridded all-satellite merged absolute dynamic topography (ADT; η) product is used to determine the importance of geostrophic current ($u_g$ denotes zonal current and $v_g$ denotes meridional current) in the GoT and the associated mechanisms:*

$$u_g = -\frac{g}{f}\frac{\partial \eta}{\partial y}, \text{ and}$$

$$v_g = \frac{g}{f}\frac{\partial \eta}{\partial x},$$

*where g represents the gravitational acceleration and f is Coriolis parameter.*

6. Line 116: Any references for reader to follow, regarding to CEOF that the author used?

    1. The author thanks the reviewer for the suggestion. A methodology section describing the complex empirical orthogonal function (CEOF) as well as providing the associated literature is added to the manuscript.

       *(L122) To determine the dominant pattern and the associated temporal variation of the surface current in the GoT, the complex empir- 110 ical orthogonal function (CEOF) is utilized. The CEOF is similar to the empirical orthogonal function (EOF) which is suitable for analysis of data with both spatial and temporal variation (e.g., Weare et al., 1976; North et al., 1982). The EOF technique decomposes the data that has its mean removed (X) into orthogonal EOF modes (U) that display spatial patterns. Each mode corresponds to a time series (V ) or the principal component (PC) that demonstrates temporal variation of that EOF mode; the PC identifies when and how intense each EOF occurs. The decomposition is done as follow:*

       *X=USV^T.*

       *Each EOF and PC explain different fractions of variance of the dataset (variance of the i-th mode is calculated as $\frac{S_{i,i}}{\sum_j S_{j,j}}$ ); the first EOF mode shows the most dominant pattern and the subsequent modes account for smaller fraction of the variance by the mathematical construction. When the technique is applied to vector quantities, e.g., velocity, the CEOF is often adopted, where each vector is transformed into a complex number (e.g., Kundu and Allen, 1976; Klinck, 1985). In this study, the velocity vector with the time-mean removed (u) is decomposed as*

       *u=u+iv,*

       *where u is the zonal velocity, v is the meridional velocity, and i is complex number, the resultant PC is complex where its magnitude represents temporal fluctuation of the corresponding EOF. The phase calculated as the imaginary part divided by the real part represents the direction that the EOF mode has to rotate (positive clockwise).*

7. Line 117: The first 2 modes represent just only 48%. What about other 52%?

    1. In this study, the CEOF technique decomposes the velocity vector fields into 173 orthogonal modes (i.e. 173 independent CEOF maps). By the mathematical setup, the first mode explains the biggest portion of the variance while the subsequent modes explain subsequently smaller portion. Therefore, the rest of the variance (58%) is distributed over 171 modes not being considered in the manuscript.

8.  Line 117-119: Without the knowledge background about the GoT dynamics, from fig.3a-c, how do you know that during southwest monsoon, anticyclonic circulation exists at the center of the GoT? Please clarify or give any references.

    1.  As shown in Figure 3a, there is an anticyclonic circulation in the middle of the GoT basin. To understand when this pattern occurs, the PC has to be considered. Magnitude of the PC indicates the intensity of the pattern and the phase of the PC indicates the rotation that CEOF1 has to rotate. During the southwest monsoon and fall monsoon transition, the phase of PC is 0 indicating no rotation needed. During the northeast monsoon and spring monsoon transition, the phase of PC is +/- pi indicating that the pattern shown in Figure 3a has to be reversed; this yields a cyclonic circulation in the middle of the GoT. The author has added a methodology section to describe the CEOF technique as well as how to interpret the results.

9.  Line 125-129: For CEOF2 "Negative phase means southward flow along the western boundary and vice versa for positive phase" How do you know that? What figures tell you that?

    1.  The original text is unclear regarding how to interpret results from the CEOF analysis, the author thanks the reviewer for the suggestion. The author has included text describing how the CEOF and PC are interpreted in the methodology section.

10. Line 164: Why strong mismatch occurs at UGoT and western boundary?

    1.  The difference between the total and geostrophic flow is simply the ageostrophic flow, i.e., flow driven by wind of affected by friction, etc. In this case, the author speculates that the large difference is likely due to strong total and geostrophic currents at these locations. Therefore, the author has rephrased the sentences to clarify this point.

        *(L220) The root-mean-square (rms) difference between the total and the geostrophic current (i.e. the estimated ageostrophic current) ranges from 0.04 to 0.11 m s−1 with the largest difference observed along the western boundary of the GoT; however, the rms difference is reasonably proportional to the speed of the total current there.*

11. Line 239: What is the source of SST? The author didn't mention it in Datasets Section.

    1.  The ENSO and IOD indices are added in the dataset section. The data sources are provided in the data availability section.

        *(L116) The weekly sea surface temperature averaged over the Niño 3.4 box (hereafter referred to as Niño3.4) provided by the National Oceanic and Atmospheric Administration (NOAA) is used to indicates ENSO conditions (Trenberth, 1997). To assess the IOD conditions, the Dipole Mode Index (DMI) is used. The weekly DMI based on sea surface temperature in the tropical Indian Ocean is calculated and provided by the NOAA/ Earth System Research Laboratory (Saji et al., 1999; Black et al., 2003).*

        *(L450) Niño 3.4 and DMI are available at https://www.cpc.ncep.noaa.gov/ data/indices/wksst8110.for and https://stateoftheocean.osmc.noaa.gov/sur/ind/dmi.php, respectively.*

12. Line 240: Please give the reference of complex correlation that the author mentioned.

*1.* Complex correlation is introduced, and reference for the calculation is provided.

*(L187) Thus, complex correlation analysis (Kundu, 1976) between current along the western boundary and that over the GoT is performed using complex number constructed from velocity vector (u, Eq. 6) to understand the dynamics associated with the strong western boundary current particularly if it is related to the current at the southeastern entrance. The complex correlation coefficient (R) is computed as*

$$R = \frac{\mathbf{u(t)}_1 \mathbf{u(t)}_2^*}{\sqrt{(\mathbf{u(t)}_1 \mathbf{u(t)}_1^*)(\mathbf{u(t)}_2 \mathbf{u(t)}_2^*)}},$$

*where $*$ denotes complex conjugate. The resultant R is a complex number where its magnitude describes how the magnitude of the two time series covary and its phase (arctan of the imaginary component divided by the real component) describes the angle between the two vector time series in order to achieve the highest correlation.*

Minor comments:

1. Line 21 and others: "Buranatheprat" -> "Buranapratheprat"

    1. The author apologizes for the wrong information provided. The text has been updated.

2. Line 26 and others: I'm not sure how to order the citation, first name or year

    1. The citation style in this manuscript follows the Copernicus Publications LaTex Package, version 6.8, 28 March 2022.

3. Line 160: Better to start new paragraph after (Kubryakov …..)

    1. The text has been updated accordingly.

4. Lines 299 & 304: "farther"

    1. The text has been updated.

5. Line 357: "Copernicus Marine Service Information"

    1. The text has been updated.

6. For the Reference, the journal name is full or short name? I found both full and short name (Be consistency).

    1. The author apologizes for the inconsistency. The reference has been updated.

7. Figure 2: unit of Fig.2a is m or m/s?

    1. The author thanks the reviewer for pointing out the mistake. The unit has to be m for (a) and $m^2$ for (b) and the corrections have been made.

[Figure]

8. Figure 3: Could not see Marron boxes in (a) and (d).

    1. The maroon boxes are missing from the figure and have now been inserted.

[Figure]

9. Figure 4: better to have a title for colorbar in the figure and have colorbar for every row.

    1. The title has been added to the figure. Since all subplots share the same colorbar, only one colorbar is provided on the lower right corner of the figure. The figure caption has been updated to emphasize this point.

        *Figure 4. The 2014 - 2019 monthly mean OSCAR current (black; left column), geostrophic current estimated from satellite altimetry (red; left column), ageostrophic flow (green; right column), and wind driven Ekman current (brown; right column) over the Gulf of Thailand in March representing the spring monsoon transition (a) and (b), June representing the southwest monsoon (c) and (d), September representing the fall monsoon transition (e) and (f), and December representing the northeast monsoon (g) and (h). The color contour shown in subplots on the left (right) column indicates magnitude of the OSCAR (ageostrophic) current with the color scale presented on the lower right corner. The scale for the current is shown on the upper right corner of (g) and (h).*

10. Figure 7: intext, author use r instead of R as shown in the figure. Just curios! When the author shows the comparison between sea surface height at different areas and negative wind stress curl, which clearly show perfect match but why r in the figure is negative not positive?

1. The author has updated the text to use R to represent the correlation coefficient.

2. The author thanks the reviewer for pointing the inconsistency. The correlation coefficient shown on the figure is from the correlation between wind stress curl and ADT; therefore, the number is negative. However, the reviewer is correct that presenting the figure and correlation coefficient this way is confusing so the figure has been updated.

[Figure]

11. Figure 8: Since color scale in the first row and second row are different, how color contour in (a) & (b) differ from the rest? Are they correlation coefficient? Better to have the title for the colorbar and have colorbar for all rows.

    1. All subplots show correlation maps; therefore, the color contours are all showing correlation coefficients but between different variables. Since the top row ((a) and (b)) is showing correlation with the velocity, complex correlation is applied. Therefore, the value is between 0 and 1 for the color contour in (a) and (b). The author understands that the figure caption might not be clear, so the text has been updated to emphasize this point.

       *Figure 8. Correlation maps between the Niño 3.4 index and OSCAR current (a), sea surface height (c), zonal wind stress (e), wind stress curl (g), and those between the Dipole Mode Index and OSCAR current (b), sea surface height (d), zonal wind stress (f), wind stress curl (h). Complex correlation was applied to the correlation between OSCAR current and the climate mode indices ((a) and (b)) with gray contour showing the phase associated with the complex correlation map.Solid (dashed) gray line is plotted every π/4 showing the positive (negative) phase relationship and thick gray line shows the zero contour. The maroon boxes in (a) and (b) designate the western boundary region (the western-most box) and the Gulf of Thailand interior region (the easternmost box). Dashed maroon boxes in (a) are areas that show high low-frequency variability used in 9c. Thick black contour in (c) – (h) denotes 0 and thin black contour is plotted every 0.25. Crosses in (c) - (h) mark the regions where the correlation is not significant with 95% confidence.*

    2. The title is added at the top of the colorbar (upper right corner of the figure).

3. Colorbars are added for all the rows.

[Figure]

12. Figure 9: Better to have no. of data legend equal to the no. of variables. For example, 9(a) has 3 lines, so it should have 3 data legend, velocity, Nino3.4 and DMI.

1. The accordant change has been made.

[Figure]

Anonymous Reviewer #2:

The paper introduces a discussion about the GoT surface circulation using remotely sensed data. The impacts of monsoon wind as well as those of ENSO and IOD on GoT surface circulation are discussed. Complex EOF and correlation analyses were used to investigate the drivers of surface circulation patterns. Seasonally, Complex EOF showed that about 28% of changes in surface circulation were attributed to the monsoon wind reversal. On interannual timescales, ENSO and IOD have spatially varying impacts on surface circulation with ENSO influence being more pronounced in the GoT interior whereas the GoT western boundary responds more to IOD conditions as evidenced by correlation analysis.

The paper is fairly well written. It introduces an important topic that can benefit a larger community to understand the impacts of climate on surface circulation and the extension to biogeochemical processes. However, there are a number of improvements that can help build a stronger case.

1. The significance of the study should be emphasised. Besides the use of field observations and numerical modelling to study the surface circulation of GoT, previous studies also used remotely sensed data but findings did not converge. So, what makes the application of remotely sensed data in this study relevant. Also, most of the remotely sensed data used in this study were available from the 90s but the analysis only focused on the period starting from 2014. Why?

    1. This study improves the understanding on the dynamics associated with the GoT circulation that has not been well-studied. It shows the role of wind on modifying both geostrophic and ageostrophic current. In addition, the study provides the variability of the circulation at the interannual timescale. The author tried to emphasize the significance of this study better in abstract, results, and conclusion sections.

    2. Thanks to the reviewer for the helpful suggestion, discussions of the interaction between GoT circulation and (1) the upper GoT and (2) the SCS are also added to the manuscripts.

        *(L262) Although the spatial resolution of the OSCAR products is too coarse to capture the complex circulation over such small and shallow basin as the uGoT, previous studies have described the general circulation (Yanagi et al., 2001; Buranapratheprat et al., 2006, 2008; Saramul and Ezer, 2014) and thus speculation on the interaction between the uGoT and the rest of the GoT can be provided. During the southwest monsoon, southward flow is present along the western boundary to the south of ~12.5° N (Figure 4c) and the uGoT circulation is anticyclonic; thus, the flow along the western boundary is likely not continuous into the uGoT and a divergence likely occurs at the coast. Instead, the northward flow along the western boundary of the uGoT is likely supplied by the western flank of the anticyclonic circulation in the GoT interior transporting high salinity (Buranapratheprat et al., 2002) and low nutrient (Buranapratheprat et al., 2009) water into the uGoT. The current at the southeast of the uGoT that is eastward/ northeastward opposing the southward flow along the uGoT eastern boundary yielding a convergence. During the northeast monsoon, cyclonic circulation is present over the uGoT. The southward flow along the uGoT western boundary potentially continues southward joining the western flank of the cyclonic circulation in the GoT interior (Figure 4g). Along the uGoT eastern boundary, the northward current is supplied by water from lower latitudes with relatively high salinity and low-nutrient (Buranapratheprat et al., 2002, 2009). The water mass is likely*

*derived from the westward/ northwestward flow between 11° and 12° N that bifurcates to have one flank travels northward along the uGoT eastern boundary while the other continues westward serving as the northern flank of the GoT interior cyclonic eddy.*

*(L309) Since the GoT connects to the SCS, variability of the SCS circulation would provide a better understanding on the GoT circulation as well as origin of water masses transported into the basin. In the southern part of the SCS, the circulation is highly influenced by the monsoon winds (e.g. Hu et al., 2000; Gan et al., 2006). During the northeast monsoon when the inflow from the SCS to the GoT is observed (Figure 4g), a strong southwestward flow is present off the eastern coast of Vietnam; the current partly turns northwestward transporting water into the GoT (Hu et al., 2000; Gan et al., 2006; Liu et al., 2008). Thus, the GoT largely is replenished by water from the northern SCS which is highly influenced by the Kuroshio intrusion (Chao et al., 1996; Jilan, 2004; Wang et al., 2006; Centurioni et al., 2009). As the Kuroshio intrusion path can be quite variable, the northern SCS circulation varies depending to the intrusion path (Hu et al., 2000; Caruso et al., 2006; Nan et al., 2015) which potentially contributes to variability of the GoT circulation, particularly at the entrance. During the southwest monsoon, a northwestward flow is present to the south of the GoT and the current off Vietnam coast reverses to flow northwestward (Hu et al., 2000; Gan et al., 2006; Liu et al., 2008). The observed GoT outflows during the southwest monsoon and fall monsoon transition likely join the northwestward flow transporting freshwater from river runoffs (Aschariyaphotha and Wongwises, 2012) into the SCS.*

3. The reviewer is correct that most of the datasets are available since the 90s but under a time constrain of the provided grant, the author can only analyze these few years of data. The author picked 2014 – 2019 period as it includes a few major interannual events, such as the 2015/2016 El Nino, the 2016 negative IOD, 2017/2018 La Nina, and the 2019 positive IOD.

2. In addition to datasets section, please also add a methods section. Most of the statistical analysis used were neither thoroughly explained nor proper references were provided. This is a weakness of the current manuscript. Complex correlation, rms correlation, etc. are not common correlation analysis methods and should be well introduced with proper citation prior to their use. Without such information it is difficult for the reader to make an easy interpretation of results presented. As for the datasets, please include the data source URLs in the datasets section and explanation about the ENSO and IOD data, and their provenance.

1. The author thanks the reviewer for the suggestion. The methodology section is added to describe the complex empirical orthogonal function.

*(L122) To determine the dominant pattern and the associated temporal variation of the surface current in the GoT, the complex empir- 110 ical orthogonal function (CEOF) is utilized. The CEOF is similar to the empirical orthogonal function (EOF) which is suitable for analysis of data with both spatial and temporal variation (e.g., Weare et al., 1976; North et al., 1982). The EOF technique decomposes the data that has its mean removed (X) into orthogonal EOF modes (U) that display spatial patterns. Each mode corresponds to a time series (V ) or the principal component (PC) that demonstrates temporal variation of that EOF mode; the PC identifies when and how intense each EOF occurs. The decomposition is done as follow:*

*$X=USV^T$.*

*Each EOF and PC explain different fractions of variance of the dataset (variance of the i-th mode is calculated as $\frac{S_{i,i}}{\sum_j S_{j,j}}$); the first EOF mode shows the most dominant pattern and the subsequent modes account for smaller fraction of the variance by the mathematical construction. When the technique is applied to vector quantities, e.g., velocity, the CEOF is often adopted, where each vector is transformed into a complex number (e.g., Kundu and Allen, 1976; Klinck, 1985). In this study, the velocity vector with the time-mean removed (u) is decomposed as*

*$u=u+iv$,*

*where u is the zonal velocity, v is the meridional velocity, and i is complex number, the resultant PC is complex where its magnitude represents temporal fluctuation of the corresponding EOF. The phase calculated as the imaginary part divided by the real part represents the direction that the EOF mode has to rotate (positive clockwise).*

2. Complex correlation is introduced, and reference for the calculation is provided.

   *(L187) Thus, complex correlation analysis (Kundu, 1976) between current along the western boundary and that over the GoT is performed using complex number constructed from velocity vector (u, Eq. 6) to understand the dynamics associated with the strong western boundary current particularly if it is related to the current at the southeastern entrance. The complex correlation coefficient (R) is computed as*

$$R = \frac{\mathbf{u(t)}_1 \mathbf{u(t)}_2^*}{\sqrt{(\mathbf{u(t)}_1 \mathbf{u(t)}_1^*)(\mathbf{u(t)}_2 \mathbf{u(t)}_2^*)}},$$

   *where $*$ denotes complex conjugate. The resultant R is a complex number where its magnitude describes how the magnitude of the two time series covary and its phase (arctan of the imaginary component divided by the real component) describes the angle between the two vector time series in order to achieve the highest correlation.*

3. The ENSO and IOD indices are added in the dataset section. The data sources are provided in the data availability section.

   *(L116) The weekly sea surface temperature averaged over the Niño 3.4 box (hereafter referred to as Niño3.4) provided by the National Oceanic and Atmospheric Administration (NOAA) is used to indicates ENSO conditions (Trenberth, 1997). To assess the IOD conditions, the Dipole Mode Index (DMI) is used. The weekly DMI based on sea surface temperature in the tropical Indian Ocean is calculated and provided by the NOAA/ Earth System Research Laboratory (Saji et al., 1999; Black et al., 2003).*

   *(L450) Niño 3.4 and DMI are available at https://www.cpc.ncep.noaa.gov/ data/indices/wksst8110.for and https://stateoftheocean.osmc.noaa.gov/sur/ind/dmi.php, respectively.*

4. OSCAR and ADT data with spatial resolution of 27–37 km were used to discuss surface circulation patterns in the uGoT. This is a small (~ 100 km horizontally) and shallow area (Figure 1). The author recognises the limitations (L340) but still places a lot of emphasis on the variability of OSCAR current data in the uGoT (e.g., L120, L135, L184, etc.). I would recommend introducing the L340 text earlier and limit the discussion of uGoT surface circulation based on OSCAR data. Please note that the circulation patterns in the uGoT have been discussed using local wind data derived from meteorological stations as a way to overcome the limitations of coarse resolution remotely sensed wind data (Buranapratheprat et al., 2006).

   1. The author thanks the reviewer for the suggestion. Discussion regarding the uGoT circulation, particularly that derived from the OSCAR products, has been limited and the author introduced the limitation regarding the observations over the uGoT in the dataset section (L87) and again at the beginning of the results (Circulation in the Gulf of Thailand) section (L146).

      *(L87) The difference between OSCAR velocity and high-frequency coastal-radar velocity is the largest in the uGoT; as the region is quite small and shallow (Figure 1), the spatial resolution provided by the OSCAR products might not be sufficient to resolve the circulation there.*

      *(L146) The mean circulation pattern from OSCAR products generally agrees with the satellite-derived geostrophic current (color contour in Figure 2a), except in the uGoT where OSCAR products are present at only six locations. Also, as the uGoT is shallow and enclosed by land on the western, northern, and eastern sides (Figure 1), OSCAR products over the region could contain substantial error. Thus, discussion regarding OSCAR velocity over the uGoT will be omitted. High variance of the surface circulation is found along the western boundary of the GoT, approximately between 9.5° and 11.5° N, with most of the variance associated with meridional velocity (Figure 2b).*

   2. The introduction has been updated and the reference has been added.

      *(L36) A study based on numerical simulation forced by spatially varying wind (Buranapratheprat et al., 2006) emphasizes the importance of both zonal and meridional wind gradient on the circulation over the uGoT. For example, the development of an anticyclonic circulation to the north of 13° N during the southwest monsoon, observed by Buranapratheprat et al. (2002) and Saramul and Ezer (2014), is highly dependent on the intensity of wind at the south or east of the uGoT.*

3. The sequence of some figures should be revised to make the flow easier. Complex EOF (Figure 3) can be understood with easy after Figure 4 is introduced, and after the method has also been introduced. Similarly, the correlation maps in Figure 8 are better after Figure 9 is introduced which will be in line with the text in L250.

   1. The author used the CEOF to understand the dominant pattern associated with circulation over the GoT. The CEOF analysis finds that the most distinct circulation pattern associates with the seasonal monsoon winds, thus, the monthly circulation is examined in detail. Therefore, the author believes that the original figure sequence provides the more coherent story.

   2. The methodology is added immediately after the dataset section.

   3. Similarly, the author used the correlation map to understand which regions are heavily impact by each climate mode. Then, the regions with high impact, i.e., that along the western boundary and that in the GoT interior with 2 sub-regions, are further analyzed to understand how low-frequency variability over

these regions change with the climate modes. The author believes that presenting figure 9 before showing figure 8 would cause confusions to the readers.

Details

L30: please revise the text for clarity. Numerical simulations were done for uGoT but the spatially uniform wind does not represent that of GoT?

The original text pointed out that the study by Yanagi and Takao (1998) indicates that the wind field over the entire GoT (including the uGoT), and thus the use of uniform wind field over the uGoT could yield inaccurate results. The author understands that the original text might provide a misleading detail, and thanks the reviewer for pointing it out. Thus, the text has been updated.

*(L34) Still, both numerical simulations are forced by spatially uniform reanalysis wind products, which is likely not representing the actual wind field over the region (Yanagi and Takao, 1998).*

L38: "The study suggests an overall cyclonic circulation…" Which study?

The author refers to the study by Saramul (2017) which is based on the high spatial-resolution coastal radar mentioned in the previous sentence. The text has been updated to clarify this point.

*(L43) The circulation based on the coastal radar suggests an overall cyclonic circulation in the northern part of the uGoT (north of 12.8°-12.9° N) and anticyclonic circulation in the southern part (south of 12.8°-12.9° N) during both monsoon seasons.*

L55: better open a new paragraph.

The text has been updated.

L77: It is helpful to add the locations of the high frequency radar system in Figure 1.

Figure 1 has been updated to include locations of the high frequency coastal radar stations.

[Figure]

L80: please explain how this comparison was done. How the spatial and temporal resolution differences were addressed? As admitted in L340 the OSCAR data may contain large error there. How to distinguish between the two, large difference or large error?

The monthly mean data as provided by Saramul et al. (2017) is calculated from the OSCAR products for the comparison; the text has been updated to clarify this point. As of the author's understanding, it is not possible to pinpoint whether the difference is truly a difference or an error. However, given the process associated with estimating the OSCAR products, the derived current along coastal areas could contain high error and a validation is necessary. As there has been no validation of OSCAR products against an observation over the GoT, the comparison to an observation which is the high frequency coastal radar is needed. The results show that the OSCAR products are reliable over the GoT, particularly to the south of the uGoT.

*(L85) To validate the OSCAR velocity over the GoT, the monthly average velocity maps in February 2015 and June 2015 are compared to tide-removed surface currents from high frequency radar system shown in Saramul (2017). Generally, OSCAR velocity exhibits similar circulation pattern to the coastal-radar velocity particularly in February 2015.*

L90: 20 km is below the ADT spatial resolution. How to know this is not noisy data?

The reviewer is correct, noisy data can contribute to a lower correlation coefficient. However, it is difficult to specify which factor contributes to the low correlation at the specific location. The author has updated the text to address this issue.

*(L103) The lower correlation found at FP could be due to the tide gauge's location,*

*which locates ~20 km inland from the available gridded satellite ADT, or the error that the ADT might have at that location.*

L99-100: references for A and wind stress curl estimation?

The estimations of A and wind stress curl are added.

*(L115)… A=8×10⁻⁵|U|²·²). In addition, wind stress curl (* $\nabla \times \tau = \frac{\partial \tau^y}{\partial x} - \frac{\partial \tau^x}{\partial y}$ *) is also calculated from the CCMPv2 product.*

L115: why use complex EOF? Does it improve the results over the classical EOF. What (additional) information is gained from the use of complex EOF? Again, these issues can be addressed by a proper methods section.

In many studies, complex EOF is applied when the dataset being considered is in vector form. The statement is included in the methodology section added to this manuscript.

L120: much of this discussion should be removed from the text. 6 grid points are too rough to have a meaningful discussion. Any derived parameter will even include less grid pixels which further limits the interpretations.

The author thanks the reviewer for the suggestion. The discussion has been removed.

L125: briefly explain the reader how to look at the results of complex EOF. For classical EOF the mode often indicates temporal variability when multiplied by the amplitude. What is the relationship between Figure 3a-c?

A brief explanation regarding how to interpret the results of complex EOF is included at the end of the added methodology section. The CEOF mode (Figure 3a) describes the dominant pattern that accounts for 28% of the total variance of the velocity anomaly over the GoT. Generally, the pattern shown in Figure 3a is observed when the phase of PC is zero and the pattern reverses when the phase is $\pm\pi$. Rotation of the CEOF pattern, specified by the phase of the corresponding PC (Figure 3c), is to be considered together with the CEOF of interest. Magnitude of the PC (Figure 3b) identifies the intensity of the pattern.

L151: reference(s) for the correlation coefficient from non-parametric method?

The reference is added to the manuscript (L198).

L165: what is rms correlation? reference(s)?

The text refers to root-mean-square of the correlation coefficient. Rms stands for root-mean-square as defined earlier (L220). To compute the rms, a square root of the mean of the squared quantity of interest (correlation coefficient over the interested area in the GoT in this case), i.e. $\sqrt{\overline{R^2}}$. The rms calculation is commonly used in similar way as calculating average; hence the reference is not necessary. Using the rms instead of the average emphasizes the importance of the variance being explained.

L171: what is total current? Most of these details should be addressed by a proper methods section.

> The author understands the confusion since the original text does not clearly address the different type of currents. A statement has been added in the datasets section to clarify this point.
>
> *(L91) Therefore, OSCAR current represents the total current (sum of the geostrophic and ageostrophic currents) over the GoT.*

L224: what is the difference between this correlation coefficient and the other so far introduced?

> The correlation here is calculated between the wind stress curl and sea surface height data, while the previous correlations (subsection 4.1.2 Geostrophic and ageostrophic component) indicate how (1) the geostrophic current and the total current covary (Figure 6a, b) and (2) the ageostrophic current and the wind-driven Ekman current (Figure 6c, d).

L227: "…much of the correlation is attributed…"

> The accordant change has been implemented.

L232: "Still, the result suggests the importance of coastal trapped Kelvin waves…" add reference(s) as this is speculative?

> Coastal trapped Kelvin waves have the special characteristic in term of the direction they travel which is based on the wave equation. They are known to travel equatorward along the western boundary of the basin and poleward along the eastern boundary. Therefore, the statement regarding the Kelvin wave traveling direction is not speculative. Still, the author agrees that an addition of a reference could be beneficial particularly to readers who are not familiar with the wave.
>
> (L302) *Still, the result suggests the importance of coastal trapped Kelvin waves which travel equatorward along the western boundary of the basin (Wang, 2002). Coastal trapped Kelvin waves are also commonly found in regions with shallow and complex bathymetry, e.g., the Indonesian Archipelago (Sprintall et al., 2000; Delman et al., 2018).*

L239: add the description (and/or sources) of the sea surface temperature and the Niño 3.4 box in the text.

> Descriptions for Niño 3.4 and DMI are added at the end of the datasets section and the corresponding sources are added to the data availability statement.
>
> *(L116) The weekly sea surface temperature averaged over the Niño 3.4 box (hereafter referred to as Niño3.4) provided by the National Oceanic and Atmospheric Administration (NOAA) is used to indicates ENSO conditions (Trenberth, 1997). To assess the IOD conditions, the Dipole Mode Index (DMI) is used. The weekly DMI based on sea surface temperature in the tropical Indian Ocean is calculated and provided by the NOAA/ Earth System Research Laboratory (Saji et al., 1999; Black et al., 2003).*

*(L450) Niño 3.4 and DMI are available at https://www.cpc.ncep.noaa.gov/ data/indices/wksst8110.for and https://stateoftheocean.osmc.noaa.gov/sur/ind/dmi.php, respectively.*

L245: as mentioned above, the discussion in this paragraph suggested that the sequence of Figure 8 and 9 is reversed.

The author used the correlation map to understand which regions are heavily impact by each climate mode. Then, the regions with high impact, i.e., that along the western boundary and that in the GoT interior with 2 sub-regions, are further analyzed to understand how low-frequency variability over these regions change with the climate modes. The author believes that the original figure sequence provides the more coherent story. Reversing the order of figure 8 and 9 could confuse the reader regarding the region selected to compare with each climate mode.

L250: "…correlations between low-frequency Niño3.4 and selected forcings…". What parameter is the forcing, Niño3.4 or ADT, etc.?

The forcings are ADT, zonal wind stress, and wind stress curl addressed in Line 344. The author investigated the impact of these forcings on the low-frequency variability of the uGoT circulation.

L300: "…but might relate to the winter warm pool" where?

The author thanks the reviewer for pointing out the missing detail. The accordant change has been made.

*(L376) The mechanism setting up the low-frequency ADT variability is unclear; it cannot be explained by local wind stress and wind stress curl (Figure 8e, g) but might relate to the winter warm pool located at the eastern boundary of the GoT (Li et al., 2014).*

L310: the last sentence is long, break it into smaller parts.

The accordant change has been made.

*(L386) However, it explains less than 5% of the low-frequency zonal current variance and the general circulation pattern away from the western boundary and the uGoT do not show a significant deviation from the seasonal current during the negative IOD event in 2016 and positive IOD event in 2019 (Figure 8b, 9c).*

Please consider revising the style of figure captions. I think starting with the label followed by the explanation is a commonly used caption style and is easy to read. E.g., Figure 1: Map of the Gulf of Thailand. (a) Triangles indicate the locations of tide gauges: FP denotes Fort Phrachula Chomklao station, KL denotes Ko Lak station, and KM denotes Ko Mattaphon station. (b) Shows the location of the Gulf of Thailand. Colour contours in (a) and (b) represent bathymetry. The black contour in (a) represents the zero-depth level.

The author thanks the reviewer for the suggestion. The author tried the suggested caption style but cannot successfully change all of them without creating awkward sentences. Therefore, the author kindly asks the reviewer to allow leaving the caption format as is.

Figure 3. I could not locate boxes in Figure 2.

The boxes are missing from the original figure; the figure is corrected.

[Figure]

Figure 5. Please use the positive half of the (b) Figure colour palette. On (a) blue corresponds to extreme positive but on (b) the same blue is extreme negative. This can be very confusing.

The author has changed the color contour to avoid the mentioned confusion.

[Figure]

Figure 6. Same as in Figure 5. Add zero contours in (b) and (d).

The author has changed the color contour and the zero contours are added in (b) and (d).

[Figure]

Figure 7. sea surface height (black) just needs to be mentioned once. How can a negative wind stress curl have both negative and positive values? I think the point here is that the displayed data was multiplied by (-1) to be in phase with the other parameter? Winds act on a much larger scale, so I am not clear about the point of the correlation between the green cross and triangle.

1. The figure caption has been updated accordingly.

   *Figure 7. Comparison between sea surface height anomaly (black) and wind stress curl (colors): both sea surface height anomaly and wind stress curl (orange) averaged over the entire Gulf of Thailand (a), sea surface height anomaly at the Gulf of Thailand western boundary shown as purple cross in Figure 2b and wind stress curl (purple) to the south of the upper Gulf of Thailand shown as purple triangle in Figure 2b (b), and both sea surface height anomaly and wind stress curl (green) to the south of the upper Gulf of Thailand shown as green cross and triangle in Figure 2b (c). Correlation coefficient between each comparison is shown on the upper right corner of each subplot. Note the reversed y-axis for wind stress curl.*

2. The figure has been updated to plot wind stress curl on the second (right) y-axis with the reversed y-axis.

[Figure]

3. In many scenarios, winds act on large scale, such as the seasonal current reversal that responds to the monsoon wind as being discussed in this manuscript and change in the warm water volume in the western equatorial Pacific due to the westerly wind bursts related to the development of ENSO events (e.g. Chen et al., 2016). However, winds do not necessarily have large-scale impact on the ocean current, for example, upwelling response to the local alongshore wind. This process is both local and small-scale. Dynamics, particularly those associated with planetary waves, associated with the ocean circulation are also sometimes remote and not providing basin-scale effects, for example, the impact of wind stress curl in the western Pacific can propagate westward in forms of Rossby waves and modify sea level at particular location in the Solomon Sea (Anutaliya et al., 2019). The effect of remotely-force waves on sea level or thermocline variation is commonly found (e.g. Dickinson, 1968; Sprintall et al., 2000; Delman et al., 2016). The author understands that impacts of planetary waves on circulation in the GoT has not been hitherto studied and could be unfamiliar to the readers. Therefore, the author has updated the text to provide some literacy regarding the topic.

*(L293) Since an effect of winds on the ocean circulation is not necessarily local nor applied over a large scale (e.g. Meyers, 1996; Giddings and MacCready, 2017), relationship between ADT at the selected locations and wind stress curl over the entire GoT is examined to identify the location of wind stress curl that influences the ADT.*

*(L303) Coastal trapped Kelvin waves are also commonly found in regions with shallow and complex bathymetry, e.g., the Indonesian Archipelago (Sprintall et al., 2000; Delman et al., 2018) and the SCS and the East China Sea (Wang et al., 2003; Yin et al., 2014; Liu et al., 2011).*

Reference:
Anutaliya, A., Send, U., Sprintall, J., McClean, J. L., Lankhorst, M., & Koelling, J. (2019). Mooring and seafloor pressure end point measurements at the southern entrance of the Solomon Sea: Subseasonal to interannual flow variability. Journal of Geophysical Research: Oceans, 124, 5085– 5104. https://doi.org/10.1029/2019JC015157

Chen, S., Wu, R., Chen, W., Yu, B. and Cao, X. (2016), Genesis of westerly

wind bursts over the equatorial western Pacific during the onset of the strong 2015–2016 El Niño. Atmos. Sci. Lett., 17: 384-391. https://doi.org/10.1002/asl.669

Dickinson, R. E. (1968). Planetary Rossby Waves Propagating Vertically Through Weak Westerly Wind Wave Guides, Journal of Atmospheric Sciences, 25(6), 984-1002.

Figure 8. Non-significant correlation could be masked?

The figure has been updated accordingly.

[Figure]

Reference

Buranapratheprat A, Yanagi T, Sojisuporn P, Booncherm C (2006) Influence of local wind field on seasonal circulation in the Upper Gulf of Thailand. Coast Mar Sci 30(1):19–26

The author thanks the reviewer for the reference.

The paper presented the surface circulation in the Gulf of Thailand from remotely sensed observations. There are many interesting study results presented in this paper. I propose to accept this paper for publication with minor revision.

The comments are

The GoT is influenced by the SCS. More results related to the influence from the SCS should be mentioned or analyzed.

> More discussions regarding the interaction with the SCS is added in both the introduction, results, and conclusion sections.
>
> *(L18) Although the GoT circulation also heavily dependent on in- flows from the SCS, e.g., along the eastern coast of Malaysia and around the southern coast of Vietnam, these currents are also mainly driven by the monsoon winds (e.g. Wyrtki, 1961; Akhir, 2012).*
>
> *(L203) During the spring monsoon transition (represented by March), the current resembles that during the northeast monsoon; a westward flow at the southeastern entrance is observed. The cyclonic circulation in the GoT interior is still present, but weak. However, the northward flow along the western boundary is stronger and wider compared to that during the northeast monsoon; width of the northward current reaches ~100° E (Figure 4a, d). Similarly, the GoT circulation in September, representing the fall monsoon transition, resembles that during the southwest monsoon despite the weak anticyclonic circulation in the interior. The circulation pattern during the monsoon transitions shows the dominant circulation pattern captured by both CEOF1 and CEOF2 (Figure 3) reflecting the influence of both monsoon winds and the current connecting to the SCS (Figure 5).*
>
> *(L309) Since the GoT connects to the SCS, variability of the SCS circulation would provide a better understanding on the GoT circulation as well as origin of water masses transported into the basin. In the southern part of the SCS, the circulation is highly influenced by the monsoon winds (e.g. Hu et al., 2000; Gan et al., 2006). During the northeast monsoon when the inflow from the SCS to the GoT is observed (Figure 4g), a strong southwestward flow is present off the eastern coast of Vietnam; the current partly turns northwestward transporting water into the GoT (Hu et al., 2000; Gan et al., 2006; Liu et al., 2008). Thus, the GoT largely is replenished by water from the northern SCS which is highly influenced by the Kuroshio intrusion (Chao et al., 1996; Jilan, 2004; Wang et al., 2006; Centurioni et al., 2009). As the Kuroshio intrusion path can be quite variable, the northern SCS circulation varies depending to the intrusion path (Hu et al., 2000; Caruso et al., 2006; Nan et al., 2015) which potentially contributes to variability of the GoT circulation, particularly at the entrance. During the southwest monsoon, a northwestward flow is present to the south of the GoT and the current off Vietnam coast reverses to flow northwestward (Hu et al., 2000; Gan et al., 2006; Liu et al., 2008). The observed GoT outflows during the southwest monsoon and fall monsoon transition likely join the northwestward flow transporting freshwater from river runoffs (Aschariyaphotha and Wongwises, 2012) into the SCS.*
>
> *(L369) This results in a weak GoT inflow at the southeastern entrance during an El Niño event consistent with weak circulation found in the SCS (Chao et al., 1996; Wang et al., 2006).*

*(L435) During the fall monsoon transition, southward flow is present along the western boundary and strong southeastward current is observed at the southeastern entrance. As the western boundary current is connected to that at the southeastern entrance (Figure 5), the results highlight the connection between circulation in the GoT and the SCS that distinctly occurs during the monsoon transitions. Moreover, variability of the circulation during the transition seasons could largely impact the properties of water in the GoT.*

The uGoT region used in this paper should be clearly defined because it is mentioned differently in the paper.

*The author thanks the reviewer for pointing this out; the text has been updated to be consistent.*

*(L15) The Gulf of Thailand (GoT) located at 8°-14° N, 99° - 105° E (Figure 1) is a shallow semi-enclosed basin with an average depth of 40 m that is largely influenced by winds on both seasonal and interannual timescales.*

CEOF is widely used in the atmospheric and marine scientific research and is also used in the paper, but it is rarely found to be used to study for the GoT. Therefore the CEOF method should be briefly described, although it is well known.

*The author thanks the reviewer for the suggestion. The methodology section is added to describe the complex empirical orthogonal function.*

*(L122) To determine the dominant pattern and the associated temporal variation of the surface current in the GoT, the complex empir- 110 ical orthogonal function (CEOF) is utilized. The CEOF is similar to the empirical orthogonal function (EOF) which is suitable for analysis of data with both spatial and temporal variation (e.g., Weare et al., 1976; North et al., 1982). The EOF technique decomposes the data that has its mean removed (X) into orthogonal EOF modes (U) that display spatial patterns. Each mode corresponds to a time series (V) or the principal component (PC) that demonstrates temporal variation of that EOF mode; the PC identifies when and how intense each EOF occurs. The decomposition is done as follow:*

*$X=USV^T$.*

*Each EOF and PC explain different fractions of variance of the dataset (variance of the i-th mode is calculated as $\frac{S_{i,i}}{\sum_j S_{j,j}}$ ); the first EOF mode shows the most dominant pattern and the subsequent modes account for smaller fraction of the variance by the mathematical construction. When the technique is applied to vector quantities, e.g., velocity, the CEOF is often adopted, where each vector is transformed into a complex number (e.g., Kundu and Allen, 1976; Klinck, 1985). In this study, the velocity vector with the time-mean removed (u) is decomposed as*

*u=u+iv,*

*where u is the zonal velocity, v is the meridional velocity, and i is complex number, the resultant PC is complex where its magnitude represents temporal fluctuation of the corresponding EOF. The phase calculated as the imaginary part divided by the real part represents the direction that the EOF mode has to rotate (positive clockwise).*

There is an error in Figure 1, please revise.

The author cannot locate the error and would greatly appreciate it if the reviewer could provide any detail regarding the error.

Please review the references, for examaple: line 370, Atoms?

The author apologizes for the wrong information provided. The text has been revised and the reference has been reviewed.

The validations of the results were still unclear. Please describe the results of the main goal clearly.

This study improves the understanding on the dynamics associated with the GoT circulation that has not been well-studied. It shows the role of wind on modifying both geostrophic and ageostrophic current. In addition, the study provides the variability of the circulation at the interannual timescale. The author tried to emphasize the significance of this study better in both abstract (L1), results (L174), and conclusion sections (L404).

---

## Author Response (AR2)

**Author Responses**
**Surface circulation in the Gulf of Thailand from remotely sensed observations: seasonal and interannual timescales**
Arachaporn Anutaliya

The author comment is presented in the following sequence: (1) itemized comments from the editor in black, (2) author's response in red, (3) the revised text is in red italic.
Thank to the editor and the reviewer for your helpful comments.

**Comments to the author**:

Editor comments (as already sent to you)
The author was not aware of the comments and had missed them. The author apologizes for not taking care of these comment earlier.

I agree with the referee comment that figure captions would be clearer if panel labels (a), (b) etc. came before the description of their content. It may help that figure captions do not have to be complete sentences (verbs are not required).
As one referee commented, some of your sentences are very long and the manuscript could be easier to read if you divided long sentences into two or three shorter ones. If you want to link the parts, you can use ";" rather than "." between the two or three parts.
The author is sincerely thankful to both the editor and the reviewer for your suggestions to improve the manuscript. The author tried the new caption format and breaking down long sentences.

Line 159. In relation to Reviewer 1 comment 7, you have not said why you ignore CEOF modes 3, 4, . . . This would be strange if mode 3 accounted for 13.9% of the variance. An empirical criterion is given by the trend in variance accounted for by modes 173, 172, . . . 5, 4, 3, 2, 1. Which of the modes 1, 2, 3, . . . clearly account for more variance than would be estimated by extrapolating the trend from the higher modes?
The first few modes of the CEOF computed from the OSCAR circulation accounts for 28%, 14%, 10%, 7.4%, etc. The text has been updated to clarify this point.

(L169) *The first few CEOF modes explain 28%, 14%, 10%, and 7.4% of the surface current variance, respectively. Therefore, only the first 2 modes are considered to represent the dominating GoT circulation patterns (Figure 3).*

Line 162. "the seasons." Maybe "these seasons" or state which seasons.
The accordant change has been implemented.

Line 210. "both" tends to suggest the winds in two monsoon states and not necessarily the current. Maybe ". . influence of monsoon winds and of the current . ." if you want to include the current as an influence.
The author thanks the editor for pointing this out. The change has been made to make the sentence clearer.

Figure 8 caption lines 5, 6. I think the boxes are blue, not maroon. [I think you cite figure 8 before figure 9 – and many times before the second mention of figure 9 – so their order seems logical.]
The figure caption has been updated accordingly.

**Referee comments**

The manuscript is responsive to the criticism raised by the reviewers of the initial submission.

I suggest the equation (7) be moved to methodology.

Equation (7) has been moved to methodology.

In response to this reviewer the author stated "the author tried to emphasize the significance of this study better in abstract…". I would suggest adding a closing sentence that summarises the main results and their implications.

A sentence has been added to highlight the main finding of the study.

(L13) *The results highlight the complex circulation pattern as being contributed by different dynamics over each region of the GoT.*

The new version mentions eastern coast of Malaysia (L28) but there is not indication of where Malaysia is located in Figure 1.

The author thanks the reviewer for bringing up this point. As Malaysian northern border is ~6.7° N, Figure 1 does not include Malaysia. However, the text has been updated to mention the location of Malaysia.

(L20) *Although the GoT circulation also heavily dependent on inflows from the SCS, e.g., along the eastern coast of Malaysia located to the south of ~6.7° N and around the southern coast of Vietnam, these currents are also mainly driven by the monsoon winds (e.g. Wyrtki, 1961; Akhir, 2012).*

Quality of Figure 3 could be improved.

Figure 3 has been modified: lines are thickened and dots are enlarged. The figure is also enlarged.

The manuscript needs editing to correct errors in grammar, language usage, clarity, etc. [Editor note: "minor revision" means that I will look at it again; moreover, it will be copy-edited by the publisher Copernicus]

The text has been updated to correct as much language errors to the best of the author's ability.

---

## Author Response (AR3)

**Author Responses**
**Surface circulation in the Gulf of Thailand from remotely sensed observations: seasonal and interannual timescales**
Arachaporn Anutaliya

The author's comment is presented in the following sequence: (1) itemized comments from the editor in black, (2) author's response in red, (3) the revised text in red italic.

Thank to the editor for your comments.

**Comments to the author**:

Detailed comments
You use "high," "higher" and "highly" too much in my opinion. Eg in Abstract line 2, better "much wind variability . ."; line 5 ". . the stronger influence . ."; line 11, omit "highly" because "dominates" makes the point. Some other cases are below but you might like to look through yourself.

The author appreciates the suggestion to improve the text. The accordant changes have been made.

Line 32. Better ". . suggests an overall cyclonic circulation . ."

The text has been updated accordingly.

Line 41. Better ". . 2008). Fine-spatial-resolution . ."

The author thanks for the suggestion. The change has been made.

Line 52. "westward current to the south of the uGoT" is not clear in figure 1 where the "cyclonic circulation" only shows in the "southward current within 1° of the GoT western boundary".

The schematic drawn to the south of 12.5° N in Figure 1 shows the results of this study to demonstrate how the circulation pattern to the south of 12.5° N connects to that in the uGoT. Although the study shows the presence of the "southward current within 1° of the GoT western boundary" consistent with that by Sojisuporn et al. (2010), the "westward current to the south of the uGoT" is not observed. Therefore, the "westward current to the south of the uGoT" is not presented in Figure 1. The author realizes that the original figure caption might not be clear, so it has been updated. Clarification is also added in the conclusions section. (L439)

*Figure 1. Map of the Gulf of Thailand (GoT). (a) Schematic of the surface current to the south of ~12.5° N suggested by this study (thick line) and that to the north of ~12.5° N derived from previous studies (e.g., Yanagi et al., 2001; Buranapratheprat et al., 2006, 2008; Saramul and Ezer, 2014, thin line) during the southwest monsoon (orange) and the northeast monsoon (blue).*

*(L439) Although the anticyclonic geostrophic circulation at the rim of the GoT (Sojisuporn et al., 2010) is not distinct, the southward geostrophic flow along the western boundary is strong.*

Lines 63-64. "dominance of an anticyclonic circulation at the rim of the GoT" is not clear in figure 1.

Similar to above, the "anticyclonic circulation" is not found in this study and thus it is not presented in Figure 1.

Line 88. "higher" –> "finer".

The text has been updated.

Line 102. "high" –> "strong".
  The accordant change has been made.

Line 104. "lower" –> "weaker".
  The text has been updated.

Line 106. "an increasing trend" –> "a rise". I don't think you want to say that the trend is increasing (meaning changing), only that it is positive.
  The author thanks the editor for pointing this out. The editor is correct; the trend is positive but not increasing. The accordant change has been implemented.

Equation (5). Please explain S.
  A sentence is added to explain *S*.

  *(L135) S contains the magnitude of a linear transformation; it designates the intensity of each EOF mode.*

Line 143. Please insert "," after "phase" and after "real part".
  The author thanks the editor for the correction, and the text has been updated.

Line 161. "High" –> "Large"
  The accordant change has been made.

Line 170-171. These "explained" variances suggest that maybe even mode 2 may be "noise" rather than a meaningful pattern. It does not rise much about the trend set by mode 4 and 3 variances. Anyway, I agree that two modes are enough to study.
  The author also had a difficult time associating patterns shown by mode 3 and mode 4 with physical forcings.

Line 183. Omit first "The".
  The text had been updated.

Lines 196-197. "(Figure 4c,d)."
  The author thanks the editor for the correction; the text has been updated accordingly.

Line 245. "compared" –> "compare"
  The accordant change has been made.

Line 261. "Higher" –> "Stronger"
  The author thanks the editor for the word choice suggestion; the text has been updated.

Line 262. "highest" –> "largest"
  The text has been updated.

Line 267. Better ". . Thus, non-zero phase relationships hint at significance of forcings . ."?
  The author thanks the editor for the suggestion; the text has been updated.

Line 339. "highly" –> "strongly"
  The accordant change has been made.

Line 343. "high" –> "strong"
  The text has been updated.

Line 361. "of up to" –> "as large as" (twice)
        The author appreciates the suggestion; the change has been implemented.

Line 373. "high" –> "strong"
        The text has been updated.

Line 379. "high" –> "strong"
        The text has been updated.

Line 422. ". . the observations reveal . ."
        The author thanks the editor for the correction; the change has been made.

Lines 433-434. "reversing circulation pattern at the surface following the monsoon wind reversal that accounts for 28%". If you mean that the . . . pattern accounts for 28% then you want ", . . . ," or "( . . . )" around "following the monsoon wind reversal". At present you are saying that the reversal accounts for 28% - not quite the same thing.
        The author intended to say that the pattern accounts for 28% and so the sentence has been updated accordingly. The author thanks the editor for your help to clarify the sentence.

Lines 445-446. Similar to lines 433-434; at present you are saying that transitions show the presence of strong seasonal meridional flow. More precisely I think you want ". . GoT circulation during monsoon transitions; there is strong seasonal meridional flow . ."
        The sentence has been updated.

Lines 447-448. Better ". . a strong northward flow along the western boundary is superimposed on the circulation pattern during the northeast monsoon (Figure 3-4). . ."?
        The author thanks the editor for the suggestion. The sentence has been updated.

Figure 8 caption line 6. "high" –> "much" or "large"
        The text has been updated.